# BideDPO: Conditional Image Generation with Simultaneous Text and Condition Alignment

**Dewei Zhou**[1], **Mingwei Li**[1,4], **Zongxin Yang**[2], **Yu Lu**[1],
**Yunqiu Xu**[1], **Zhizhong Wang**[3], **Zeyi Huang**[3], **Yi Yang**[1,*]

[1]Zhejiang University    [2]Harvard University    [3]Central Media Technology Institute, Huawei
[4]Zhongguancun Academy, Beijing

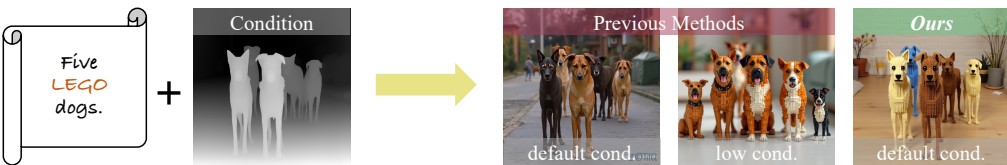

(a) **Input-Level Conflict: Text** requires LEGO structure vs. **Condition** image requires real-dog structure.

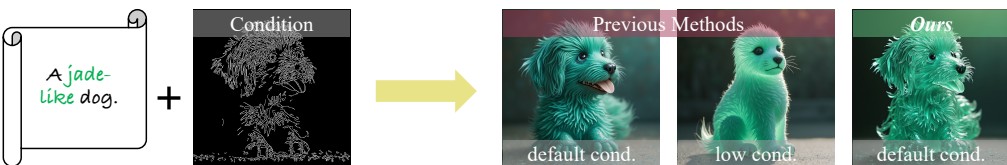

(b) **Model-Bias Conflict: Text** requires jade-like surface vs. **Model-Bias** favors real-dog texture.

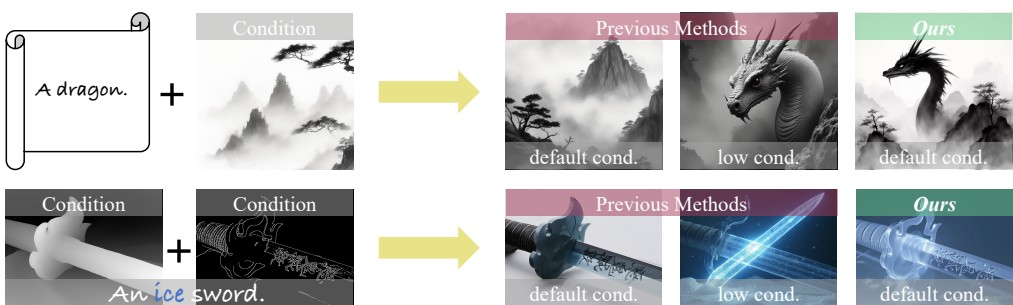

(c) **More examples achieving dual alignment between text prompts and conditioning inputs.**

Figure 1: **Qualitative comparison on cases with conflicting text and condition.** We first introduce two conflicts between the text prompt and conditioning input: (a) Input Level Conflict and (b) Model Bias Conflict, which hinder model controllability. We then propose a solution that resolves both, generating images that satisfy both the text and the condition. "default cond." means using its default condition constraint scale, while "low cond." means using a lower condition constraint scale. (c) Our method also enhances the alignment between text and abstract conditions such as style condition, and supports generation with multiple conditions combined with text prompts.

## ABSTRACT

Conditional image generation augments text-to-image synthesis with structural, spatial, or stylistic priors and is used in many domains. However, current methods struggle to harmonize guidance from both sources when conflicts arise: 1) input-level conflict, where the semantics of the conditioning image contradict the text prompt, and 2) model-bias conflict, where learned generative biases hinder alignment even when the condition and text are compatible. These scenarios demand nuanced, case-by-case trade-offs that standard supervised fine-tuning struggles

---

*Corresponding author.

to deliver. Preference-based optimization techniques, such as Direct Preference Optimization (DPO), offer a promising solution but remain limited: naive DPO suffers from gradient entanglement between text and condition signals and lacks disentangled, conflict-aware training data for multi-constraint tasks. To overcome these issues, we propose a self-driven, bidirectionally decoupled DPO framework (BideDPO). At its core, our method constructs two disentangled preference pairs for each sample—one for the condition and one for the text—to mitigate gradient entanglement. The influence of these pairs is then managed by an Adaptive Loss Balancing strategy for balanced optimization. To generate these pairs, we introduce an automated data pipeline that iteratively samples from the model and uses vision-language model checks to create disentangled, conflict-aware data. Finally, this entire process is embedded within an iterative optimization strategy that progressively refines both the model and the data. We construct a DualAlign benchmark to evaluate a model's ability to resolve conflicts between text and condition, and experiments on commonly used modalities show that BideDPO delivers substantial gains in both text success rate (e.g., +35%) and condition adherence. We also validated the robustness of our approach on the widely used COCO dataset. Project Pages: `https://limuloo.github.io/BideDPO/`.

# 1 INTRODUCTION

Conditional image generation (Zhang et al., 2023; Li et al., 2024; Liu et al., 2024; Zavadski et al., 2024) augments text-to-image synthesis with auxiliary constraints (*e.g.*, structural or spatial priors) and is now widely used in digital art, design, and related workflows. However, real-world use with complex prompt–condition pairs reveals a fundamental yet underexplored challenge: reconciling guidance from the text prompt with the conditioning input. **In this work, we are the first to explicitly identify this problem and propose a solution.** Specifically, we highlight two recurrent conflicts that undermine model controllability: **1) Input-Level Conflict.** When the condition image contains strong semantics that contradict the user prompt, current models often fail to balance these competing sources of guidance. As shown in Fig. 1(a), under the official default setting, models typically prioritize the condition image, resulting in outputs that closely replicate its semantics while neglecting the prompt. Conversely, weakening the influence of the condition allows the model to better follow the text, but often at the expense of spatial or structural consistency. **2) Model-Bias Conflict.** Modern conditional generation models possess strong generative bias—that is, given a particular condition input, the model tends to produce outputs consistent with its learned biases. As illustrated in Fig. 1(b), even when the condition and text are theoretically compatible, a mismatch between the model's prior and the prompt can lead to poor adherence to the textual guidance. Addressing the above conflicts requires the model to effectively navigate trade-offs between the condition input and the text prompt, for which no universally optimal solution exists. In this context, rather than directly providing the model with fixed "correct" outputs through supervised learning, a more flexible and effective alternative is to guide the model using preference data—examples that reflect human judgments over competing outputs. Motivated by this, we adopt the Direct Preference Optimization (DPO) (Rafailov et al., 2023) approach, which has been shown to effectively align model outputs with human preferences in both large language models and standard text-to-image generation, and apply it to train our model to better resolve conflicts between condition inputs and text prompts based on preference data.

However, introducing DPO into the image-conditioned generation task proves to be highly nontrivial, presenting two major challenges. **1) Naive DPO fails to achieve balanced alignment of both constraints.** In naive DPO, a single preference pair is used for each example. To jointly improve both condition and text alignment for each case, it is necessary to set the positive sample to satisfy both constraints and the negative sample to satisfy neither (Fig. 2(a)). However, we observe that the model often prioritizes the condition input while neglecting the text, especially when these guidance signals conflict (see Fig. 5). This limitation stems from gradient entanglement, as the coupled learning signals obscure the optimization direction for the weaker constraint, making it difficult for the model to balance and improve both constraints simultaneously. **2) Lack of Disentangled, Conflict-Aware Preference DPO Data:** To the best of our knowledge, there are no established DPO datasets tailored for conditional image generation, especially for scenarios where the condition input

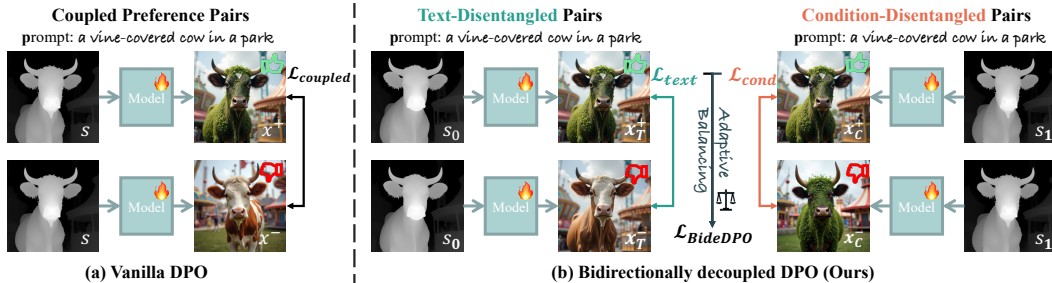

Figure 2: **Comparison between vanilla DPO and our bidirectionally decoupled DPO for conditional image generation.** a) Vanilla DPO uses coupled preference pairs, so its gradients can become ambiguous or even vanish when text and condition are not aligned together. b) BideDPO separates the learning signals for text and condition and adaptively balance them. This provides clear, adaptive gradients for each requirement, allowing the model to achieve better multi-constraint alignment.

and text prompt provide conflicting or competing guidance. This data gap significantly limits the exploration and benchmarking of preference-based optimization in multi-constraint settings.

To overcome these challenges, we propose a **self-driven, bidirectionally decoupled DPO framework** tailored for image-conditioned generation. Our framework consists of three key components:
**1) Bidirectionally decoupled DPO algorithm (BideDPO)**: As shown in Fig. 2(b), unlike naive DPO, our method constructs two decoupled preference pairs per example—one for condition fidelity and one for text adherence—used simultaneously during optimization.

An adaptive loss balancing strategy further ensures balanced progress on both objectives, preventing the model from collapsing toward a single constraint. This decoupling, combined with dynamic loss adjustment, enables the model to better handle conflicts between the two constraints and achieve a more effective trade-off. **2) Automated Construction of Disentangled and Conflict-Aware DPO Preference Data**: As shown in Fig. 3, we address the lack of suitable DPO data by introducing an automated pipeline that iteratively samples from the current model and uses vision-language model (VLM) checks to construct high-quality positive and negative samples for both text and condition branches. These datasets include numerous instances where the text and the condition are in conflict. **3) An iterative optimization strategy**: Our framework naturally supports iterative refinement because the generator itself produces the preference data used for training. As shown in Fig. 4, we alternate between generating preference pairs with the current model and optimizing it with BideDPO. Each round leverages the improved generator to produce higher-quality data, creating a self-reinforcing loop that progressively enhances both model performance and data quality.

We construct a DualAlign benchmark to evaluate how state-of-the-art conditional image generation methods handle conflicts between the conditioning input and the text prompt. Experiments are conducted across standard conditioning modalities. Results show that our method markedly improves the text success rate (SR) and adherence to the conditioning signal over strong baselines—for example, on FLUX-Depth it boosts text SR by **15**% and reduces the conditional MSE by **87.7** points. We also validate robustness on the standard COCO benchmark (Lin et al., 2015; Zhang et al., 2025): even with standard prompts, our approach delivers substantial gains over the original model, e.g., a **15**% improvement for canny-conditioned generation. Furthermore, our iterative optimization strategy proves effective: as the number of iterations increases, the model consistently achieves better trade-offs between conditioning and text, yielding steady performance gains.

We summarize our contributions as follows:

- To the best of our knowledge, this work is the first to formally formulate and systematically analyze the text–condition adherence conflict in conditional image generation. We propose BideDPO, an effective DPO algorithm that reconciles conflicts between condition inputs and text prompts.

- We propose an automated pipeline that produces disentangled condition–text preference pairs. It easily extends to other tasks and supports iterative optimization, improving both model and data.

- We construct a DualAlign Benchmark for evaluating a model's ability to handle conflicts between condition inputs and text prompts, and the results highlight the effectiveness of our approach.

## 2 RELATED WORK

**Conditional Image Generation.** With the rapid advancement of image generation technology (Rombach et al., 2022; Podell et al., 2023; Rombach et al., 2023; Esser et al., 2024; Lu et al., 2023; 2024a; 2025; 2024b; Zhou et al., 2025c; 2023; 2024a;b;c; 2025b;a; Zhao et al., 2024a;c; 2025a;b; 2024b; Li et al., 2025; 2026), current models are now capable of producing highly realistic images. As a result, a key research focus has shifted toward enabling more conditional image generation (Li et al., 2024; Bhat et al., 2024; Lin et al., 2024; Peng et al., 2024; Wang et al., 2024; Ye et al., 2023; Zhang et al., 2025; Xu et al., 2025), aiming to make text-to-image technologies applicable in a broader range of real-world scenarios. For example, models of the ControlNet (Zhang et al., 2023; Xu et al., 2024; Zhao et al., 2023) family, such as FLUX.1-dev-UnionPro2 (Shakker Labs, 2025), introduce an auxiliary network to inject user-provided spatial constraints or reference image information into the generation process. Other variants, including FLUX-Depth and FLUX-Canny (BlackForest, 2024), concatenate the encoded condition image with the generated image features at the input channel, training the entire network end-to-end to achieve conditional image generation.

**Aligning Image Generation with Human Preferences.** Reinforcement learning (RL) methods are widely used for post-training large language models (LLMs) to align outputs with human preferences, and has recently been applied to image generation for improved controllability and preference alignment. Most approaches rely on explicit reward models, such as ImageReward (Xu et al., 2023), combined with policy rollouts like PPO (Liu et al., 2025; Xue et al., 2025) or direct gradient methods (Clark et al., 2024; Prabhudesai et al., 2023). A more efficient alternative is Direct Preference Optimization (DPO) (Rafailov et al., 2023), adapted to diffusion models by Diffusion-DPO (Wallace et al., 2023), and further extended for richer feedback (RankDPO (Karthik et al., 2024)) and timestep inconsistencies (SPO (Liang et al., 2024), TailorPO (Ren et al., 2025)). Given its efficiency and effectiveness, we build on the DPO framework. However, existing DPO-based methods still struggle with conditional image generation involving multiple, potentially conflicting constraints. To address this, we propose a novel **Bidirectionally Decoupled DPO** algorithm that enables clearer optimization directions and better handles complex conditional scenarios.

## 3 METHOD

Our method has three components: (1) a Bidirectionally Decoupled DPO algorithm (BideDPO, §3.1) that enforces simultaneous text–condition adherence; (2) an automatic pipeline for constructing Disentangled and Conflict-Aware Preference Data (§3.2) for BideDPO training; and (3) an iterative optimization strategy (§3.3) that jointly improves model performance and data quality. **Preliminary background on Diffusion Models and DPO is provided in Appendix §A.**

### 3.1 BIDIRECTIONALLY DECOUPLED DPO

**Limitations of Vanilla DPO.** In conditional image generation, models are often tasked with satisfying both a text prompt $p$ and an extra structural condition $s$. For notational simplicity, we encapsulate both inputs into a single composite context variable $c = (p, s)$. To analyze the optimization dynamics of DPO (Rafailov et al., 2023) in this multi-objective setting, we can conceptualize the model's preference score as being composed of a text alignment component, $f_{\text{text}}(x, c; \theta)$ (which primarily depends on $p$), and a condition alignment component, $f_{\text{cond}}(x, c; \theta)$ (which primarily depends on $s$).

For simplicity, let us assume the overall score $f(x, c; \theta)$ can be modeled as a weighted linear combination of these components, while acknowledging that the true relationship is far more complex:

$$f(x, c; \theta) = \lambda_{\text{text}} f_{\text{text}}(x, c; \theta) + \lambda_{\text{cond}} f_{\text{cond}}(x, c; \theta), \tag{1}$$

where $\lambda_{\text{text}}$ and $\lambda_{\text{cond}}$ are scalar weights. For a preference triplet $(x^+, x^-, c)$, where both the preferred and dispreferred samples are evaluated under the same shared composite context $c$, the vanilla DPO loss is:

$$\mathcal{L}_{\text{coupled}} = -\log \sigma(f(x^+, c; \theta) - f(x^-, c; \theta)). \tag{2}$$

The partial derivative of this loss with respect to the model parameters $\theta$ is given by:

$$\frac{\partial \mathcal{L}_{\text{coupled}}}{\partial \theta} = -(1 - \sigma(\Delta(c; \theta)) \frac{\partial \Delta(c; \theta)}{\partial \theta}, \tag{3}$$

$$\Delta(c;\theta) = \lambda_{\text{text}} \underbrace{\left(f_{\text{text}}(x^+,c;\theta) - f_{\text{text}}(x^-,c;\theta)\right)}_{\Delta_{\text{text}}(c;\theta)} + \lambda_{\text{cond}} \underbrace{\left(f_{\text{cond}}(x^+,c;\theta) - f_{\text{cond}}(x^-,c;\theta)\right)}_{\Delta_{\text{cond}}(c;\theta)}. \quad (4)$$

The update gradient in Eq. 3 is influenced simultaneously by both objectives, but when one objective's gradient is significantly stronger, it can dominate the update. Consequently, the weaker objective may be masked—or, in cases of conflict, the update may even move in a direction that opposes the weaker objective, making its optimization more difficult.

**Decoupled Preference Pairs.** To address this limitation, we construct two decoupled preference pairs for each case, targeting condition and text alignment separately (Fig. 2(b)). For text alignment, we use a pair $(x_T^+, x_T^-, c_0)$, and for condition alignment, we use $(x_C^+, x_C^-, c_1)$. Here, $c_0 = (p, s_0)$ and $c_1 = (p, s_1)$, where $p$ is the same target prompt in both cases. In the text alignment pair, $x_T^+$ and $x_T^-$ have similar adherence to $s_0$, but $x_T^+$ follows the prompt $p$, while $x_T^-$ does not. In the condition alignment pair, both $x_C^+$ and $x_C^-$ follow $p$, but $x_C^+$ matches $s_1$ much better than $x_C^-$. Two independent loss terms are then calculated separately for each objective:

$$\mathcal{L}_{\text{text}} = -\log \sigma \underbrace{\left(f_{\text{text}}(x_T^+, c_0; \theta) - f_{\text{text}}(x_T^-, c_0; \theta)\right)}_{\Delta_{\text{text}}(c_0;\theta)}, \quad (5)$$

$$\mathcal{L}_{\text{cond}} = -\log \sigma \underbrace{\left(f_{\text{cond}}(x_C^+, c_1; \theta) - f_{\text{cond}}(x_C^-, c_1; \theta)\right)}_{\Delta_{\text{cond}}(c_1;\theta)} \quad (6)$$

**Adaptive Loss Balancing.** To prevent the optimization from being dominated by one objective, we introduce an adaptive loss balancing strategy. To ensure stable training, the weights for each loss component are computed based on their current magnitudes but are treated as detached constants during backpropagation. This is achieved by applying a stop-gradient operator, denoted as $\text{sg}(\cdot)$:

$$w_{\text{text}} = \text{sg}\left(\frac{\mathcal{L}_{\text{text}}}{\mathcal{L}_{\text{text}} + \mathcal{L}_{\text{cond}}}\right), \quad \text{and} \quad w_{\text{cond}} = \text{sg}\left(1 - w_{\text{text}}\right). \quad (7)$$

The total loss is thus a dynamically weighted sum:

$$\mathcal{L}_{\text{decoupled}} = w_{\text{text}}\mathcal{L}_{\text{text}} + w_{\text{cond}}\mathcal{L}_{\text{cond}}. \quad (8)$$

**Decoupled gradient.** The gradient can be formulated as:

$$\frac{\partial \mathcal{L}_{\text{decoupled}}}{\partial \theta} = w_{\text{text}} \frac{\partial \mathcal{L}_{\text{text}}}{\partial \theta} + w_{\text{cond}} \frac{\partial \mathcal{L}_{\text{cond}}}{\partial \theta}$$
$$= -w_{\text{text}}\left(1 - \sigma\left(\Delta_{\text{text}}(c_0;\theta)\right)\right)\frac{\partial \Delta_{\text{text}}(c_0;\theta)}{\partial \theta} - w_{\text{cond}}\left(1 - \sigma\left(\Delta_{\text{cond}}(c_1;\theta)\right)\right)\frac{\partial \Delta_{\text{cond}}(c_1;\theta)}{\partial \theta}. \quad (9)$$

Crucially, this gradient is a fully decoupled sum. Unlike the coupled gradient in Eq. 3, our approach provides a distinct optimization signal for each objective. This prevents one objective's gradient from being diminished or "swallowed" when the other's loss is significantly larger, ensuring both are consistently optimized.

**BideDPO Objective for Diffusion Models.** Building upon prior work (Wallace et al., 2023), we define the reward $r(x_t, c, \epsilon; \theta)$ in diffusion models as the reduction in denoising error for a noisy sample $x_t$ under a given context $c$, which is computed as the difference between the denoising error $\|\epsilon - \epsilon_{\text{ref}}(x_t, c)\|^2$ of the frozen reference network and the error $\|\epsilon - \epsilon_\theta(x_t, c)\|^2$ of the optimized network, where $\epsilon$ represents the noise:

$$r(x_t, c, \epsilon; \theta) = \|\epsilon - \epsilon_{\text{ref}}(x_t, c)\|^2 - \|\epsilon - \epsilon_\theta(x_t, c)\|^2. \quad (10)$$

The total reward differences for the text pair under context $c_0$ ($R_T$) and the context pair under context $c_1$ ($R_C$) are then:

$$R_T = r(x_{t,T}^+, c_0, \epsilon_T^+; \theta) - r(x_{t,T}^-, c_0, \epsilon_T^-; \theta), \quad (11)$$

$$R_C = r(x_{t,C}^+, c_1, \epsilon_C^+; \theta) - r(x_{t,C}^-, c_1, \epsilon_C^-; \theta). \quad (12)$$

The final BideDPO loss adaptively weights the objectives based on these reward differences:

$$\mathcal{L}_{\text{BideDPO}}(\theta) = -\mathbb{E}_{(x_T^+, x_T^-, c_0) \sim \mathcal{D}_T, (x_C^+, x_C^-, c_1) \sim \mathcal{D}_C}\left[w_{\text{text}} \log \sigma(\beta T R_T) + w_{\text{cond}} \log \sigma(\beta T R_C)\right]. \quad (13)$$

By providing a distinct gradient for each objective, our decoupled approach mitigates the interference inherent in the coupled DPO loss. This leads to more stable and efficient multi-objective optimization. **Please see Appendix §G for more details.**

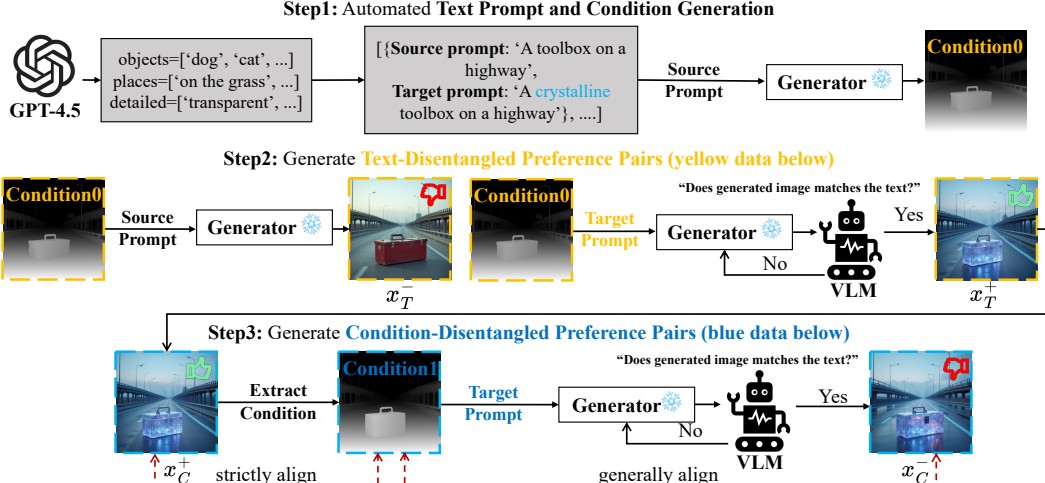

Figure 3: **The Automated Disentangled, Conflict-Aware Preference Data Generation Pipeline.**

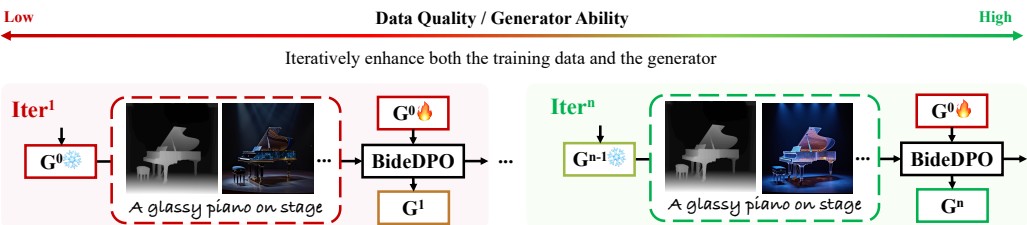

Figure 4: **Iterative Optimization Strategy.** We start with an initial generator ($G^0$) that produces training data via our automated pipeline (Fig. 3). Training with BideDPO already improves the model ($G^1$), while repeating the process with the updated generator yields higher-quality data and further gains, forming a self-reinforcing loop where both data and model improve progressively.

## 3.2 AUTOMATED CONSTRUCTION OF DISENTANGLED AND CONFLICT-AWARE DPO PREFERENCE DATA

As shown in Fig. 3, we design an automated data construction pipeline that explicitly generates disentangled preference pairs for both text and condition alignment, including cases where the two objectives are in conflict. It consists of three steps:

**1. Prompt and Initial Condition Generation.** We first use an LLM to generate a basic *Source Prompt* and a more detailed *Target Prompt* $p$. The source prompt is then used to produce an initial condition map, "Condition 0" ($s_0$). The generated $s_0$ and target prompt $p$ often exhibit input-level or model-bias conflicts in Fig. 1.

**2. Text-Disentangled Pair** ($x_T^+, x_T^-, p, s_0$)**.** Both samples adhere to "Condition 0" ($s_0$). The preferred sample $x_T^+$ is generated from the *Target Prompt*, with its textual alignment verified by a VLM, and serves as our high-quality anchor. The dispreferred sample $x_T^-$ is generated from the *Source Prompt* and thus lacks textual alignment with *Target Prompt*.

**3. Condition-Disentangled Pair** ($x_C^+, x_C^-, p, s_1$)**.** Both samples align with the *Target Prompt* $p$. The anchor $x_T^+$ serves as the preferred sample $x_C^+$, and a new, strictly aligned condition map "Condition 1" ($s_1$) is extracted from it. The dispreferred sample $x_C^-$ is then generated to adhere less strictly to "Condition 1" while matching the target prompt's semantics.

This structured process allows us to systematically generate preference data that isolates and targets distinct aspects of text and condition alignment. By repeating this process over a large set of prompts and conditions, we built a comprehensive dataset that enables targeted, disentangled optimization for multi-constraint image generation. Moreover, this approach can be easily extended to other tasks, such as style-text alignment, as illustrated in Fig. 8.

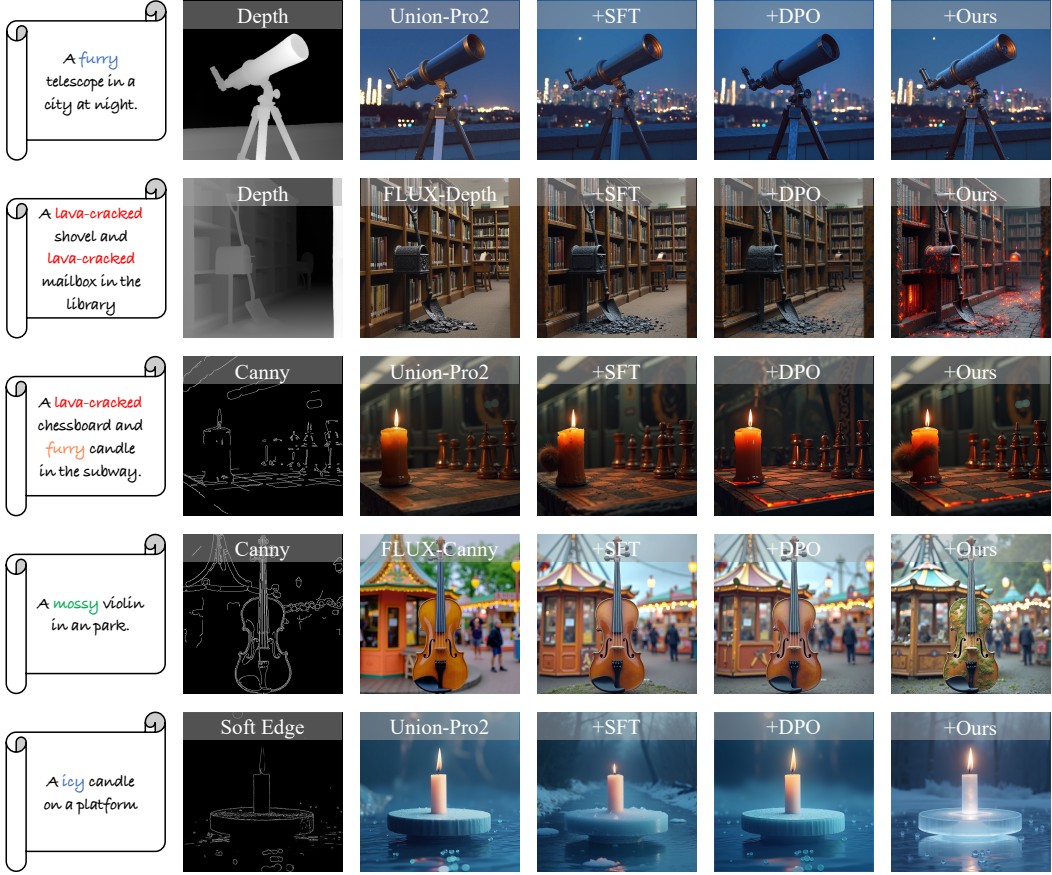

Figure 5: **Visual comparison for conditional image generation on the DualAlign Benchmark.** We evaluate three common conditioning modalities: depth, Canny, and soft edge. Our method improves adherence to both the text prompt and the spatial conditioning. **Please zoom in for details.**

### 3.3 ITERATIVE OPTIMIZATION STRATEGY

For each generator model, BideDPO strengthens adherence to both text and condition, and since our data construction pipeline builds samples directly from the same model, the process naturally supports iterative refinement. As shown in Fig. 4, we alternate between generating preference data with the current model and optimizing it with BideDPO, forming a self-reinforcing loop that progressively improves both the generator and its training data.

## 4 EXPERIMENTS

### 4.1 EXPERIMENTAL SETUP

**Baselines.** We conduct experiments on the state-of-the-art text-to-image model FLUX (BlackForest, 2024). Specifically, we evaluate our approach on the most widely used conditional image generation variants in the community, including FLUX-Depth, FLUX-Canny, and Union-Pro2 (Shakker Labs, 2025). We also compare with LooseControl (limited to depth conditioning) (Bhat et al., 2024) and ControlNet++ (Li et al., 2024). We evaluate style-conditioned generation on FLUX IP-Adapter (Team, 2024). We primarily compare our method with two common baselines: supervised fine-tuning (SFT) and naive DPO (Wallace et al., 2023).

**Implementation Details.** We generate 5,000 samples in each iteration. For SFT, we use the positive samples in Fig. 3. For DPO, we construct coupled preference pairs by combining the condition image and positive sample from the Condition-Disentangled Preference Pairs, together with the negative sample from the Text-Disentangled Preference Pairs. We fine-tune all models using Low-Rank Adaptation (LoRA (Hu et al., 2022)) method with rank of 256. For the SFT method, we train

Table 1: **Results for depth-conditioned image generation on DualAlign Benchmark.** "Ctrl." indicates support for conditional generation.

| Method | Ctrl. | SR ↑ | MSE ↓ | SGMSE ↓ | CLIP ↑ |
|---|---|---|---|---|---|
| FLUX | × | 0.79 | N/A | N/A | 0.2936 |
| LooseControl | ✓ | 0.43 | 791.1 | 1280.2 | 0.2852 |
| ControlNet++ | ✓ | 0.49 | 331.9 | 480.7 | 0.2854 |
| Union-Pro2 | ✓ | 0.49 | 177.0 | 272.4 | 0.2748 |
| + SFT | ✓ | 0.70 | 262.2 | 332.5 | 0.2915 |
| + DPO | ✓ | 0.71 | 168.3 | 219.9 | 0.2860 |
| + Ours | ✓ | **0.84** | **164.0** | **195.7** | **0.2924** |
| FLUX-Depth | ✓ | 0.76 | 233.6 | 282.8 | 0.2899 |
| + SFT | ✓ | 0.79 | 162.2 | 195.7 | 0.2926 |
| + DPO | ✓ | 0.89 | 171.9 | 195.0 | 0.2974 |
| + Ours | ✓ | **0.91** | **145.9** | **164.4** | **0.2982** |

Table 2: **Results for canny-conditioned image generation on DualAlign Benchmark.** "Ctrl." indicates support for conditional generation.

| Method | Ctrl. | SR ↑ | F1 ↑ | SGF1 ↑ | CLIP ↑ |
|---|---|---|---|---|---|
| FLUX | × | 0.71 | N/A | N/A | 0.2965 |
| ControlNet++ | ✓ | 0.40 | 0.437 | 0.174 | 0.2828 |
| Union-Pro2 | ✓ | 0.34 | 0.418 | 0.143 | 0.2753 |
| + SFT | ✓ | 0.58 | 0.324 | 0.178 | 0.2838 |
| + DPO | ✓ | 0.50 | 0.607 | 0.284 | 0.2840 |
| + Ours | ✓ | **0.68** | **0.607** | **0.393** | **0.2845** |
| FLUX-Canny | ✓ | 0.33 | 0.397 | 0.129 | 0.2703 |
| + SFT | ✓ | 0.52 | 0.357 | 0.179 | 0.2842 |
| + DPO | ✓ | 0.55 | 0.452 | 0.248 | 0.2829 |
| + Ours | ✓ | **0.73** | **0.454** | **0.333** | **0.2927** |

for 5,000 steps using the Prodigy (Mishchenko & Defazio, 2023) optimizer with a learning rate of 1.0, using all positive samples in Fig. 3; for DPO and BideDPO, starting from the SFT-tuned model, we optimize for an additional 2,000 steps using the AdamW (Kingma & Ba, 2017) optimizer with a learning rate of 0.00004 and a weight decay of 0.01.

**Evaluation Benchmarks.** *1) DualAlign benchmark for conflicting text–condition constraints.* Currently, there is no established benchmark for evaluating conditional image generation in scenarios where the text prompt and condition image provide partially conflicting guidance. Therefore, we construct our own test set following a similar pipeline as our training data, generating text-condition pairs that require the model to make meaningful trade-offs between constraints. To better assess the generalization ability of our approach, we ensure that the objects, places, and detailed descriptions in the test set do not overlap with those in the training set. Each modality contains 100 cases. *2) COCO benchmark for robustness.* To assess robustness on a standard benchmark, we also evaluate various post-training methods alongside their baseline models on the COCO dataset (Lin et al., 2015; Zhang et al., 2025), demonstrating that our approach preserves the base model's original performance. *3) DualAlign-Style benchmark for text–style condition constraints (see § E.2 in Appendix).*

**Evaluation Metrics.** We evaluate our models using the following metrics: *1) Success Ratio:* We use the Qwen2.5-VL-72B (Bai et al., 2023) model to automatically determine whether the generated image accurately matches the text description, providing a direct measure of text-image consistency. *2) CLIP Score (Radford et al., 2021):* This metric quantifies the semantic alignment between the generated image and the input prompt, indicating how well the model captures the intended content described by the user. *3) MSE/F1 Score:* These metrics assess the degree to which the generated image conforms to the input condition (*e.g.*, spatial or structural constraints), thereby measuring conditional fidelity. *4) Semantic-Guided MSE (SGMSE), Semantic-Guided F1 (SGF1), and Semantic-Guided SSIM (SGSSIM):* To jointly evaluate textual and conditional alignment, we define SGMSE, SGF1, and SGSSIM. Each extends its standard counterpart by adding a semantic check. If a generation fails the text requirement, we apply a penalty: MSE is doubled, and the F1 or SSIM score is set to zero (otherwise the metrics reduce to the usual MSE, F1, and SSIM). This design penalizes outputs that do not satisfy both constraints and provides a more comprehensive assessment of controllable image generation.

## 4.2 EXPERIMENTAL RESULTS

**Qualitative Results.** Fig. 5 presents visual comparisons between our BideDPO and other post-training methods. Supervised fine-tuning (SFT) reduces the model's adherence to the input condition. For example, in the fifth row of Fig. 5, the shape at the top of the candle is noticeably altered. Naive DPO, due to coupled gradients, biases optimization toward condition adherence while often neglecting textual alignment, resulting in outputs that frequently fail to match the text description. In contrast, our bidirectionally decoupled and adaptively balanced approach enables the model to resolve conflicts between condition and text better, achieving a more effective trade-off and satisfying both constraints. Notably, BideDPO and vanilla SFT are trained with exactly the same set of positive examples; this controlled comparison underscores the superiority of our method.

Table 4: **Quantitative results on COCO Benchmark with depth and canny conditioning.**

| Method | Depth-conditioned | | | | Canny-conditioned | | | |
|---|---|---|---|---|---|---|---|---|
| | SR ↑ | MSE ↓ | SGMSE ↓ | CLIP ↑ | SR ↑ | F1 ↑ | SGF1 ↑ | CLIP ↑ |
| LooseControl | 0.72 | 1334.0 | 1706.0 | 0.2534 | N/A | N/A | N/A | N/A |
| ControlNet++ | 0.79 | 548.3 | 668.9 | 0.2557 | 0.71 | 0.339 | 0.245 | 0.2622 |
| Union-Pro2 | 0.83 | 297.3 | 363.5 | 0.2546 | 0.78 | 0.416 | 0.332 | 0.2539 |
| + SFT | 0.87 | 561.7 | 635.6 | 0.2575 | 0.81 | 0.271 | 0.271 | 0.2602 |
| + DPO | 0.90 | 263.4 | 278.2 | 0.2586 | 0.75 | 0.490 | 0.373 | 0.2554 |
| + Ours | **0.91** | **236.3** | **245.3** | **0.2633** | **0.83** | **0.497** | **0.392** | **0.2629** |

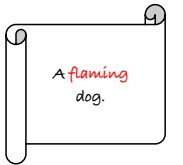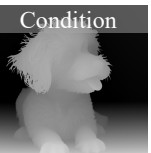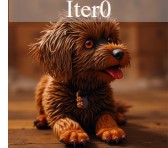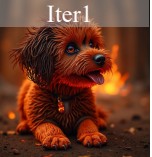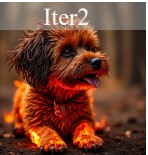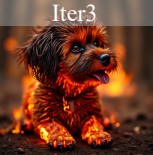

Figure 6: **Visualization of Iterative Optimization.**

**Quantitative Results.** Quantitative results in Tabs. 1, 2, and 3 show that our method substantially improves adherence to both text prompts and conditioning inputs. For example, in depth-conditioned generation, we observe a **43%** increase in Success Ratio on Union-Pro2 and a **15%** increase on FLUX-Depth. In canny-conditioned generation, our approach achieves a **34%** higher Success Ratio on Union-Pro2 and a **40%** increase on FLUX-Canny—even surpassing the original T2I FLUX model in terms of text alignment. Moreover, our method also enhances adherence to the input condition across various baselines. For instance, on the MSE loss, our approach reduces the error of FLUX-Depth by **148.687**, demonstrating improved conditional fidelity. Finally, Tab. 4 shows that our approach does not compromise the base model's robustness: on COCO—a dataset not used during training—it still delivers improvements over the original model. Importantly, all results shown here are obtained without iterative optimization.

Table 3: **Results for soft edge-conditioned image generation on DualAlign Benchmark.** "Ctrl." indicates support for conditional generation.

| Method | Ctrl. | SR ↑ | SSIM ↑ | SGSSIM ↑ | CLIP ↑ |
|---|---|---|---|---|---|
| FLUX | × | 0.73 | N/A | N/A | 0.2907 |
| Union-Pro2 | ✓ | 0.24 | 0.610 | 0.145 | 0.2768 |
| + SFT | ✓ | 0.48 | 0.510 | 0.255 | 0.2855 |
| + DPO | ✓ | 0.39 | 0.637 | 0.250 | 0.2783 |
| + Ours | ✓ | **0.49** | **0.643** | **0.297** | **0.2855** |

**Iterative optimization strategy.** Since BideDPO simultaneously strengthens the model's adherence to both text and condition, we can adopt an iterative optimization algorithm to refine the model and data together. Tab. 5 shows that our iterative optimization (Fig. 4) progressively improves adherence to both condition and text, reaching optimal performance by the third iteration. This trend is further supported by the qualitative comparisons in Fig. 6, where generated images increasingly align with the text while preserving condition fidelity. Importantly, even a single iteration already yields substantial improvements over the baseline. Thus, iterative refinement should be regarded as an optional enhancement that can be adjusted based on available computational resources.

Table 5: **Ablation study of our core components.** "w/o ALB" ablates our Adaptive Loss Balancing. "Text. Only" and "Cond. Only" are trained using only the text or condition preference pairs, respectively.

| Method | SR ↑ | MSE ↓ | SGMSE ↓ | CLIP ↑ |
|---|---|---|---|---|
| Iter = 1 | 0.84 | 163.968 | 195.728 | 0.2924 |
| Iter = 2 | 0.85 | **158.876** | 195.459 | 0.2947 |
| **Iter = 3 (Ours)** | **0.88** | 159.559 | **190.263** | **0.2957** |
| Iter = 4 | 0.86 | 166.274 | 202.363 | 0.2939 |
| w/o ALB | 0.78 | 157.729 | 205.246 | 0.2862 |
| Text. Only | 0.88 | 258.954 | 287.749 | 0.2947 |
| Cond. Only | 0.59 | 153.659 | 218.683 | 0.2753 |

**Ablation Study of Adaptive Loss Balancing (ALB).** In Tab. 5, the results for "Iter = 1" demonstrate more balanced improvements than those for "w/o ALB", with a **6%** increase in success rate and a **9.518** gain in SGMSE, despite only a modest decrease in MSE (**6.239**).

**Ablation Study on Preference Pairs.** As shown in Tab. 5, using only text- or condition-disentangled preference pairs (Fig. 3) makes the model focus on a single aspect, yielding marginal

Table 6: **User study results.** Values are win rates.

| Comparison | Text ↑ | Condition ↑ | Overall ↑ |
|---|---|---|---|
| Ours *vs.* Base | 67.9% | 66.8% | 64.0% |

Table 7: **Style-conditioned generation.**

| Method | SR ↑ | Style ↑ | SG Style ↑ | CLIP ↑ |
|---|---|---|---|---|
| IPA | 30% | **6.50** | 1.31 | 0.1679 |
| + Ours | **58%** | 6.26 | **2.97** | **0.2015** |

Table 8: **Success rates from different evaluators.** Results show that our improvements are not artifacts of a specific VLM but hold universally across models and human judge.

| Judge | UnionPro2 | +SFT | +DPO | +Ours |
|---|---|---|---|---|
| Qwen2.5 | 0.49 | 0.70 | 0.71 | **0.84** |
| GPT-4o | 0.43 | 0.65 | 0.70 | **0.82** |
| Human | 0.42 | 0.62 | 0.64 | **0.84** |

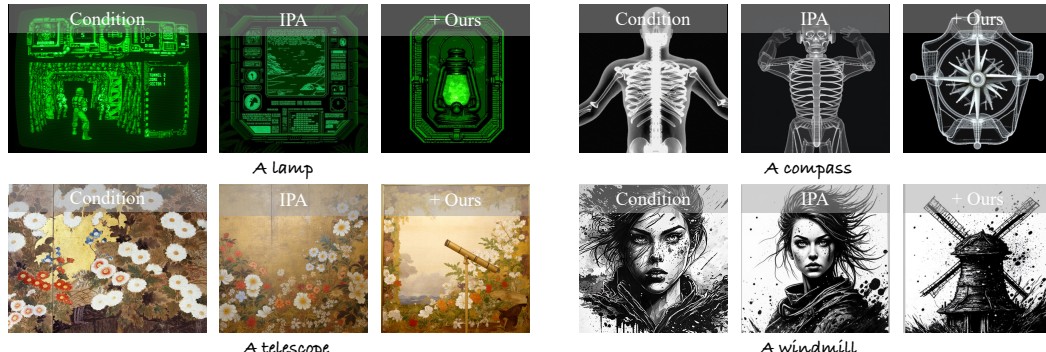

Figure 7: **Visual results for style-conditioned image generation on IP-Adapter (Team, 2024).**

or even harmful effects on the other. Only by jointly leveraging both types of preference pairs (ours) can the model achieve balanced improvements across all constraints.

**User Study.** We conducted a user study to compare all models, evaluating three aspects: text adherence, condition adherence, and overall alignment. For each trial, we randomly sampled 20 cases and asked 30 participants to evaluate them in a 1-*vs.*-1 format, yielding a total of 600 comparisons. As shown in Tab. 6, participants favored our model roughly twice as often as the baseline.

**Universality on style-conditional image generation.** Beyond structure- and spatial-conditional generation, we further validate the effectiveness of our method on more abstract conditions, such as style-conditioned image generation. As shown in Fig. 7, prior methods often suffer from excessive reference copying, due to the Model Bias issue discussed in Fig. 1(b). In contrast, our proposed BidDPO algorithm substantially mitigates this problem, achieving a 28% higher success rate (Tab. 7).

**More VLM Selection.** To verify robustness beyond a single evaluator, we additionally assessed our method using multiple VLMs—including Qwen2.5-VL-72B, GPT-4o—and human raters. As shown in Tab. 8, all evaluators yield consistent rankings, confirming that our improvements are not artifacts of a specific VLM but hold universally across models and human judgment.

## 5 CONCLUSION

In this work, we address the challenge of achieving simultaneous text and condition alignment in image generation. We identify that existing approaches—including supervised fine-tuning and naive DPO—struggle to balance multiple constraints, especially when the text prompt and condition input are in conflict. To overcome this, we propose a bidirectionally decoupled DPO framework that disentangles the optimization of textual and conditional adherence. Furthermore, adaptive loss balancing ensures stable and effective learning between objectives. Our approach also features an automated pipeline for constructing high-quality, disentangled preference pairs, as well as an iterative optimization strategy that continuously enhances both the data and the model. Experiments demonstrate that our method outperforms strong baselines on both textual and conditional alignment, yielding substantial improvements in Success Ratio and conditional fidelity across a variety of benchmarks. Our framework not only advances the state of controllable image generation, but also provides new insights into preference-based learning with multiple, potentially conflicting objectives.

**Acknowledgements.** This work was supported in part by the National Natural Science Foundation of China (62402432, 62293554, U2336212) and Zhongguancun Academy, Beijing, China (20240313).

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
