## A PRELIMINARY

**Denoising Diffusion Models.** Denoising diffusion models Ho et al. (2020) are a class of generative models that learn to synthesize data by reversing a fixed forward noising process. This process gradually adds Gaussian noise to a clean sample $x_0$ over $T$ timesteps, such that $x_t = \sqrt{\bar{\alpha}_t} x_0 + \sqrt{1 - \bar{\alpha}_t} \epsilon$, where $\epsilon \sim \mathcal{N}(0, \mathbf{I})$. A neural network $\epsilon_\theta$ is then trained to predict the added noise $\epsilon$ from the noisy sample $x_t$ timestep $t$, and context $c$. The objective is to minimize the L2 error between the actual and predicted noise:

By learning to effectively denoise at every step, the model can generate high-fidelity samples from pure noise.

$$\mathcal{L}_{\text{simple}} = \mathbb{E}_{\epsilon, t, x_0, x_t} \left[ \|\epsilon - \epsilon_\theta(x_t, t, c)\|_2^2 \right]. \tag{14}$$

**Direct Preference Optimization (DPO).** Rafailov et al. (2023) introduced DPO, a method to fine-tune LLMs with pairs of ranked examples $(x^+, x^-, c)$, where $x^+$ is the preferred and $x^-$ the dispreferred sample. The training objective is formulated using an implicit reward $\hat{r}_\theta(x, c) = \beta \cdot \log \frac{p_\theta(x|c)}{p_{\text{ref}}(x|c)}$, which measures the log-likelihood ratio with respect to a reference model $p_{\text{ref}}$ and $\beta$ is a hyperparameter. The DPO loss is expressed as:

$$\begin{aligned}
\mathcal{L}_{\text{DPO}}(\theta) &= -\mathbb{E}_{x^+, x^-, c} \left[ \log \sigma \left( \hat{r}_\theta(x^+, c) - \hat{r}_\theta(x^-, c) \right) \right] \\
&= -\mathbb{E}_{x^+, x^-, c} \left[ \log \sigma \left( \beta \left( \log \frac{p_\theta(x^+|c)}{p_{\text{ref}}(x^+|c)} - \log \frac{p_\theta(x^-|c)}{p_{\text{ref}}(x^-|c)} \right) \right) \right],
\end{aligned} \tag{15}$$

where $\sigma$ is the sigmoid function.

**Diffusion-DPO.** Wallace et al. (2023) applied DPO to diffusion models by modifying Eq. 15. They replaced the logarithmic difference by the denoising error:

$$\begin{aligned}
\mathcal{L}_{\text{D-DPO}}(\theta) = -\mathbb{E}_{x_t^+, x_t^-, c} \Big[ \log \sigma \Big( -\beta T \Big( \\
\|\epsilon^+ - \epsilon_\theta^+(x_t^+, t, c)\|_2^2 - \|\epsilon^+ - \epsilon_{\text{ref}}^+(x_t^+, t, c)\|_2^2 - \\
\|\epsilon^- - \epsilon_\theta^-(x_t^-, t, c)\|_2^2 + \|\epsilon^- - \epsilon_{\text{ref}}^-(x_t^-, t, c)\|_2^2 \Big) \Big) \Big],
\end{aligned} \tag{16}$$

where $x_t^+$ and $x_t^-$ are obtained from $x_0^+$ and $x_0^-$ using the forward process of diffusion models, and $T$ is a temperature.

## B IMPLEMENTATION DETAILS

### B.1 BASELINE MODELS

All our experiments are conducted using state-of-the-art conditional text-to-image diffusion models from the FLUX family (BlackForest, 2024). Specifically, we adopt the following publicly available, pre-trained models as our baselines:

- **FLUX-Depth:** A variant specialized for depth-conditioned image generation, implemented by the FLUX team.
- **FLUX-Canny:** A variant specialized for Canny-edge-conditioned image generation, implemented by the FLUX team.
- **Union-Pro2** (Shakker Labs, 2025): A powerful model built on top of FLUX that supports conditional generation. It incorporates the ControlNet approach by adding an extra side network to the FLUX architecture, enabling multi-modal conditioning. Union-Pro2 is one of the most widely downloaded models in the AIGC community.
- **FLUX.1-dev-IP-Adapter** (Team, 2024): An IP-Adapter implementation for FLUX.1-dev released by InstantX Team, enabling image-conditioned generation where reference images provide style conditions for generation.

We primarily compare our proposed BideDPO method against two standard baselines: Supervised Fine-Tuning (SFT) and a naive application of DPO (Wallace et al., 2023), evaluated on the models listed above.

## B.2 BASELINE CONFIGURATIONS

To ensure a fair comparison, we configured the Supervised Fine-Tuning (SFT) and naive DPO baselines as follows, using the preference data generated by our pipeline:

**SFT.** The SFT baseline was trained with a standard denoising score matching objective. We exclusively used high-quality positive samples from our generated data—specifically, the preferred samples $x_T^+$ from the text-disentangled pairs (conditioned on the target prompt $p$ and initial condition $s_0$) and $x_C^+$ from the condition-disentangled pairs (conditioned on $p$ and the refined condition $s_1$).

**Naive DPO.** For the DPO baseline, we constructed text-and-condition preference pairs to simulate a standard DPO setting in which preferences are not disentangled:

- The preferred sample $x^+$ is $x_C^+$ from the condition-disentangled pair, which aligns well with both the target prompt $p$ and the condition $s_1$.
- The dispreferred sample $x^-$ is $x_T^-$ from the text-disentangled pair. This sample is generated from the initial condition $s_0$ and does not align with the target prompt $p$, making it a poor fit for both the target prompt $p$ and the condition $s_1$.

In this way, $x_C^+$ and $x_T^-$ together form a DPO preference pair that jointly enforces adherence to both the target prompt $p$ and the condition $s_1$.

## B.3 TRAINING HYPERPARAMETERS

All models were fine-tuned on a cluster of 4 NVIDIA A100 GPUs with a batch size of 4. We employed the Low-Rank Adaptation (LoRA) (Hu et al., 2022) method with a rank of 256 for fine-tuning all models. For the SFT method, we trained for 5,000 steps using the Prodigy (Mishchenko & Defazio, 2023) optimizer with a learning rate of 1.0. For both DPO and BideDPO, starting from the SFT-tuned model, we further optimized for an additional 2,000 steps using the AdamW (Kingma & Ba, 2017) optimizer with a learning rate of 0.00004 and a weight decay of 0.01. We set the $\beta$ parameter (Wallace et al., 2023) for DPO to 5000. Each iteration (data generation + fine-tuning) uses 8×A800 GPUs (40GB) for 5 hours (data generation) and 4×A100 GPUs (80GB) for 3 hours (training).

# C EVALUATION METRIC DETAILS

In our experiments, we employ a comprehensive set of metrics to evaluate model performance across text alignment, conditional fidelity, and their combination. Below, we provide detailed descriptions of each metric.

## C.1 TEXT ALIGNMENT METRICS

### C.1.1 SUCCESS RATIO

To automatically assess whether a generated image accurately reflects the text prompt, we use the powerful Vision-Language Model (VLM) `Qwen2.5-VL-72B`. For each generated image and its corresponding target prompt, we query the VLM with a carefully designed question: "Does the image successfully depict the following description: '[Prompt]'? Please answer with 'Yes' or 'No'." The Success Ratio is then calculated as the percentage of "Yes" responses across our entire test set. This provides a direct and automated measure of text-image consistency.

### C.1.2 CLIP SCORE

The CLIP Score measures the semantic similarity between the generated image and the input text prompt. We use the pre-trained ViT-L/14 CLIP model to compute embeddings for both the image and the prompt. The score is the cosine similarity between these two embedding vectors, scaled by 100. A higher CLIP score indicates better semantic alignment with the user's textual description.

## C.2 Conditional Fidelity Metrics

### C.2.1 Mean Squared Error (MSE) and F1 Score

To quantify how well a generated image adheres to the input structural condition (*e.g.*, depth map, Canny edges), we first extract the corresponding condition map from the generated image using the same tool employed during data creation (*e.g.*, Depth Anything v2 for depth). We then compute the Mean Squared Error (MSE) or F1 Score between the extracted condition map and the original input condition map.

- **MSE:** Used for pixel-wise regression tasks like depth map prediction. A lower MSE indicates higher fidelity to the ground-truth condition.
- **F1 Score:** Used for tasks like Canny edge or human pose matching, where we can treat it as a binary segmentation problem. A higher F1 score indicates better structural correspondence.

## C.3 Combined Text and Condition Metrics

### C.3.1 Semantic-Guided MSE (SGMSE) and F1 (SGF1)

Standard conditional metrics like MSE and F1 only measure structural fidelity and ignore whether the generated image is semantically correct according to the text prompt. To address this, we introduce two novel metrics: Semantic-Guided MSE (SGMSE) and Semantic-Guided F1 (SGF1). These metrics integrate a semantic check (using the same VLM as for the Success Ratio) into the calculation:

- If the generated image is deemed a "Success" (*i.e.*, it matches the text prompt), the SGMSE and SGF1 are the same as the standard MSE and F1 scores.
- If the generated image is a "Failure" (it does not match the text prompt), we apply a penalty to reflect the semantic mismatch. The SGMSE is doubled (*i.e.*, $2 \times$ MSE), and the SGF1 score is set to zero.

This penalty mechanism ensures that the model is rewarded only when it satisfies both the textual and conditional constraints simultaneously, providing a more holistic evaluation of conditional generation.

# D Data Construction Details

## D.1 Condition Modalities

Our framework is designed to be agnostic to the specific type of conditional input. For the experiments in this paper, we constructed datasets for three different structural modalities:

- **Depth Maps:** To provide 3D scene geometry, we utilized the widely-used `Depth Anything v2` (Yang et al., 2024) model to extract high-quality depth maps from images.
- **Canny Edges:** For sharp, well-defined object boundaries, we used the standard Canny edge detection algorithm.
- **Soft Edges:** We employ the *ControlNet-SoftEdge* family of edge detectors, specifically the recent `MistoLine-SDXL` model (Lvmin Zhang, 2023).

This variety of conditions allows us to evaluate the robustness and versatility of our method across different types of structural constraints.

## D.2 More Details of Automated Data Construction Pipeline

Our data construction pipeline is designed to automatically generate disentangled preference pairs for both text and condition alignment. This process is crucial for training our model to handle multi-

objective optimization effectively, especially in cases with conflicting constraints. The pipeline, as illustrated in the main paper, consists of the following three steps:

### D.2.1 STEP 1: PROMPT AND INITIAL CONDITION GENERATION

The process begins with the generation of prompts using a large language model (LLM). For each data point, we generate a basic *Source Prompt* and a more descriptive *Target Prompt* ($p$). The source prompt is then used to create an initial, often loose, condition map, which we denote as "Condition 0" ($s_0$). This initial pairing of the target prompt $p$ and Condition 0 $s_0$ is intentionally designed to often contain conflicts, either at the input level or due to model priors, as discussed in the main paper.

### D.2.2 STEP 2: TEXT-DISENTANGLED PAIR GENERATION

With the prompts and initial condition, we generate the text-disentangled preference pair $(x_T^+, x_T^-, p, s_0)$. Both samples in this pair are generated to adhere to the same initial "Condition 0" ($s_0$).

- **Preferred Sample ($x_T^+$):** The preferred sample is generated using the detailed *Target Prompt*. Its alignment with the text is verified using a Vision-Language Model (VLM). This high-quality sample serves as a reference anchor, denoted $x_a$.

- **Dispreferred Sample ($x_T^-$):** The dispreferred sample is generated using the basic *Source Prompt*. As a result, it correctly follows "Condition 0" but lacks the specific textual details present in the target prompt, making it less preferred from a text-alignment perspective.

### D.2.3 STEP 3: CONDITION-DISENTANGLED PAIR GENERATION

Next, we construct the condition-disentangled preference pair $(x_C^+, x_C^-, p, s_1)$. For this pair, both samples are generated to align with the same *Target Prompt p*.

- **Preferred Sample ($x_C^+$):** The anchor image $x_a$ from the previous step is used as the preferred sample. A new, strictly aligned condition map, "Condition 1" ($s_1$), is then extracted directly from this anchor image.

- **Dispreferred Sample ($x_C^-$):** The dispreferred sample is generated to match the semantics of the target prompt but to adhere less strictly to the new "Condition 1". This creates a preference based on conditional fidelity. Because the generator has limited capability, the generated image $x_C^-$ will inevitably exhibit some loss of fidelity to the precise structural details of $s_1$ when compared to the original image $x_a$ from which $s_1$ was derived.

This structured, three-step process allows us to systematically generate a large dataset of preference pairs that isolate and target distinct aspects of text and condition alignment.

## E BIDEDPO ON STYLE-CONDITIONED GENERATION

### E.1 AUTOMATED STYLE-AWARE PREFERENCE PIPELINE

To incorporate explicit artistic controls into our preference data, we extend the above pipeline with the style-aware branch illustrated in Fig. 8. As shown in Step 1, GPT-5.1 enumerates object concepts together with concise style captions ("Cubist faceted planes," etc.). The LLM emits a minimalist *Source Prompt*, a descriptive *Target Prompt*, and the style caption, which is forwarded to a Web-Search API to retrieve a representative condition image.

Step 2 mirrors the step 2 of Fig. 3. Holding the retrieved condition fixed, we render a positive sample $x_{T,\text{style}}^+$ using the Target Prompt and a high IP-Adapter scale, with VLM verification ensuring that both the textual semantics and the referenced style are expressed. A negative counterpart $x_{T,\text{style}}^-$ is produced with the minimalist Source Prompt (empty string) while reusing the same condition, yielding an image that resembles the structure and style but fails to mention the target concept. A VLM adversary inspects each pair, only accepting anchors for which the text truly matches the prompt and the style hint.

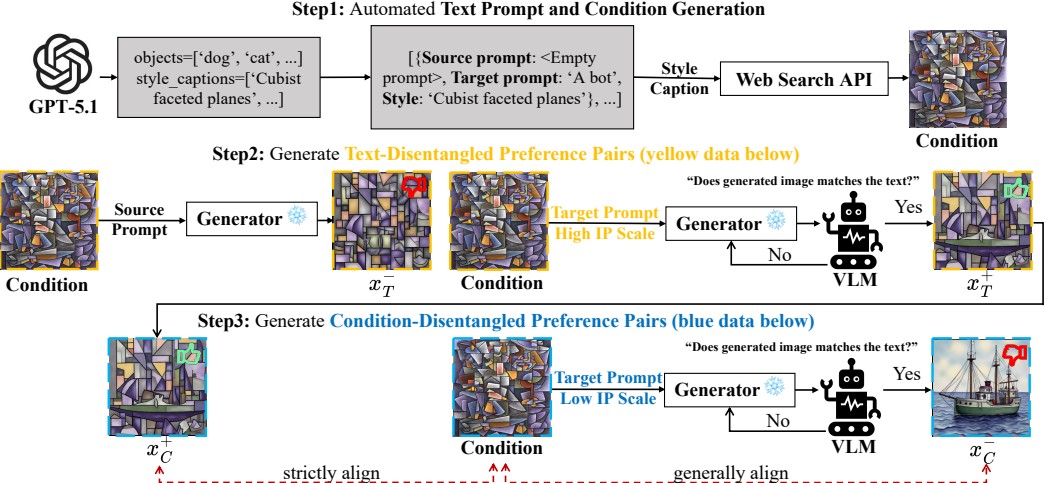

Figure 8: **The Automated Style-Aware Preference Data Pipeline.**

Step 3 then isolates conditional fidelity. We keep the Target Prompt and style condition fixed but vary the IP-Adapter scale so that $x^+_{C,\text{style}}$ strictly follows the retrieved condition map while $x^-_{C,\text{style}}$ only coarsely aligns ("strictly align" vs. "generally align" in the figure). Both samples still satisfy the textual description, so their difference arises purely from how faithfully they respect the style-conditioned control input. This yields preference pairs $(x^+_{C,\text{style}}, x^-_{C,\text{style}}, p, s_1)$ that drop seamlessly into the unified BideDPO training mix. Notably, this style-aware pipeline follows almost the same structure as our main data pipeline, highlighting the generality and versatility of our proposed preference data construction framework.

## E.2   STYLE BENCHMARK AND DATA.

Using the data generation pipeline defined in the Automated Style-Aware Preference Pipeline above, we construct a style-conditioned benchmark. The training set consists of 100 objects and 20 styles, while the test set contains 10 different objects and 10 different styles.

**Metrics.** We ask Qwen2.5-VL 72B Bai et al. (2023) to rate each generated image on a 0–10 **Style Score**, and define **SG Style Score** by zeroing the rating whenever the reference style conflicts with the Target Prompt. We also track success rate (SR)—the fraction of samples where the VLM judges both text and style as satisfied—along with CLIP similarity for semantic grounding. These metrics jointly capture textual fidelity, structural accuracy, and adherence to the curated style references.

## E.3   EXPERIMENT RESULTS

**Quantitative Results.** Tab. 7 presents the comprehensive quantitative comparison on the DualAlign-Style benchmark. As shown, BideDPO significantly improves the IP-Adapter (IPA) baseline across all key metrics. Most notably, the success rate (SR) increases from **30% to 58%**, representing a **93% relative improvement**, which demonstrates that our method substantially enhances the model's ability to simultaneously satisfy both textual semantics and style constraints. While the baseline achieves a slightly higher mean Style Score (6.50 vs. 6.26), this is expected because IPA tends to directly copy the style reference image, which naturally yields high style similarity scores but at the cost of semantic accuracy (only 30% SR). In contrast, the disentangled **SG Style Score** reveals the true capability: BideDPO achieves **2.97** compared to IPA's **1.31**, representing a **127% relative improvement**. This substantial gap demonstrates that BideDPO effectively resolves conflicts between style references and text prompts, producing outputs that maintain style fidelity even when the style condition is semantically incompatible with the target description, while IPA's high Style Score primarily reflects its tendency to copy the reference rather than harmonize style with semantics. Additionally, the CLIP score improves from 0.1679 to 0.2015, confirming better semantic alignment with the text prompts. These results demonstrate that the same bidirectionally decoupled objective extends seamlessly to abstract, reference-image style conditions without requiring

architectural modifications, simply by adding another preference head and an adaptive loss balancer weight.

**Qualitative Results.** Fig. 7 presents comprehensive qualitative comparisons on the DualAlign-Style benchmark, showcasing BideDPO's superior capability in style-conditioned generation. The visualization demonstrates four challenging generation tasks, each requiring the model to generate a specific target object (a lamp, a compass, a telescope, and a windmill) while simultaneously adhering to diverse and complex style conditions. These conditions span a wide range of artistic styles: green pixelated retro game aesthetics, X-ray transparency effects, traditional Japanese art with gold leaf accents, and gritty ink-splatter illustrations. As shown in the figure, the IP-Adapter (IPA) baseline often fails to generate the specified object, instead producing variations of the condition itself or semantically related but incorrect content. For instance, when asked to generate "A lamp" under a retro game style condition, IPA produces a green-tinted futuristic UI overlay without the target lamp object. Similarly, for "A compass" under an X-ray condition, IPA generates a human skeleton instead of the requested compass. In contrast, our BideDPO method successfully integrates the target object into the given condition's aesthetic, demonstrating superior capability in harmonizing textual semantics with stylistic constraints while maintaining both object accuracy and style fidelity. The generated images not only correctly depict the target objects but also faithfully preserve the distinctive visual characteristics of each style condition, such as the green monochrome digital aesthetic for the lamp, the transparent wireframe style for the compass, the ornate floral background with gold accents for the telescope, and the gritty ink-splatter texture for the windmill.

## F  ADDITIONAL VISUALIZATION RESULTS

### F.1  ADDITIONAL EXAMPLES OF DISENTANGLED AND CONFLICT-AWARE DPO PREFERENCE DATA

Fig. 9 presents additional examples of our Disentangled and Conflict-Aware DPO data, using depth maps to illustrate our methodology. We construct preference pairs spanning a spectrum of conditional alignment errors—large, mid-level, and minor—to train the model progressively on structural fidelity.

**Large Errors**  The top row of Fig. 9 shows pairs where the negative sample ($x_C^-$) has significant structural deviations. For instance, the generated "bejeweled barn" and "vine-covered tower" fail to match the fundamental layout of their depth maps. These examples train the model to capture the global composition.

**Mid-level Errors**  The middle row presents moderate inconsistencies. The negative samples capture the main objects but err in key aspects, such as the misplaced window in the "soot-covered window" scene or the incorrect shape of the "glowing chisel." These pairs refine the model's grasp of spatial relationships.

**Minor Errors**  The bottom row focuses on fine-grained details. The negative samples are largely faithful but contain subtle inaccuracies, such as ignoring background foliage structures in the "neon-lit penguin" scene or failing to render surface grains on the "frosted cookie." These examples hone the model's ability to render precise details, enhancing overall fidelity.

### F.2  ADDITIONAL VISUALIZATIONS OF DEPTH, CANNY, AND SOFT-EDGE CONDITIONS

In this section, we present comprehensive qualitative results that further demonstrate the superior capabilities of our proposed BideDPO method across diverse conditional image generation scenarios. Fig. 10 showcases an extensive collection of examples, systematically organized by three distinct conditional modalities, illustrating how our model achieves enhanced fidelity and semantic alignment while maintaining structural integrity across various input conditions.

#### F.2.1  DEPTH CONDITION

Fig. 10 (Top) demonstrates our method's effectiveness when conditioned on depth maps, where grayscale intensity encodes spatial distance information. The BideDPO model exhibits remarkable

---

**Algorithm 1** Automated Construction of Disentangled Preference Data

---

1: **Input:** Generator model $G$, LLM, Condition extractor $E$, VLM, Num samples to generate $N$, Max retries $K$.
2: **Initialize:** Unified preference set $\mathcal{D} \leftarrow \emptyset$.
3: *Step 1: Pre-generate prompts and initial conditions*
4: $P_{\text{pool}} \leftarrow$ LLM.generate_source_target_pairs()
5: $\text{Context}_{\text{pool}} \leftarrow \emptyset$
6: **for** $(p_{\text{source}}, p_{\text{target}})$ in $P_{\text{pool}}$ **do**
7: $\quad x_{\text{init}} \leftarrow G(p_{\text{source}})$
8: $\quad s_0 \leftarrow E(x_{\text{init}})$
9: $\quad \text{Context}_{\text{pool}} \leftarrow \text{Context}_{\text{pool}} \cup \{(p_{\text{source}}, p_{\text{target}}, s_0)\}$
10: **end for**
11: **while** $|\mathcal{D}| < N$ **do**
12: $\quad p_{\text{source}}, p_{\text{target}}, s_0 \leftarrow$ RandomSample($\text{Context}_{\text{pool}}$)
13: $\quad$ — *Step 2: Attempt to generate text-disentangled pair* —
14: $\quad x_a \leftarrow$ None
15: $\quad tries \leftarrow 0$
16: $\quad$ **while** $tries < K$ **do**
17: $\quad\quad$ candidate $\leftarrow G(p_{\text{target}}, s_0)$
18: $\quad\quad$ **if** VLM.verify(candidate, $p_{\text{target}}$) **then**
19: $\quad\quad\quad x_a \leftarrow$ candidate
20: $\quad\quad\quad$ **break**
21: $\quad\quad$ **end if**
22: $\quad\quad tries \leftarrow tries + 1$
23: $\quad$ **end while**
24: $\quad$ **if** $x_a$ is None **then**
25: $\quad\quad$ **continue** {Failed to generate a valid anchor}
26: $\quad$ **end if**
27: $\quad x_T^+ \leftarrow x_a$
28: $\quad x_T^- \leftarrow G(p_{\text{source}}, s_0)$
29: $\quad$ — *Step 3: Attempt to generate condition-disentangled pair* —
30: $\quad x_C^+ \leftarrow x_a$
31: $\quad s_1 \leftarrow E(x_a)$
32: $\quad x_C^- \leftarrow$ None
33: $\quad tries \leftarrow 0$
34: $\quad$ **while** $tries < K$ **do**
35: $\quad\quad$ candidate $\leftarrow G(p_{\text{target}}, s_1)$
36: $\quad\quad$ **if** VLM.verify(candidate, $p_{\text{target}}$) **then**
37: $\quad\quad\quad x_C^- \leftarrow$ candidate
38: $\quad\quad\quad$ **break**
39: $\quad\quad$ **end if**
40: $\quad\quad tries \leftarrow tries + 1$
41: $\quad$ **end while**
42: $\quad$ **if** $x_C^-$ is None **then**
43: $\quad\quad$ **continue** {Failed to generate a valid counterpart}
44: $\quad$ **end if**
45: $\quad$ — *Both pairs successfully generated, add to unified dataset* —
46: $\quad c_0 \leftarrow (p_{\text{target}}, s_0)$
47: $\quad c_1 \leftarrow (p_{\text{target}}, s_1)$
48: $\quad pair_T \leftarrow (x_T^+, x_T^-, c_0)$
49: $\quad pair_C \leftarrow (x_C^+, x_C^-, c_1)$
50: $\quad \mathcal{D} \leftarrow \mathcal{D} \cup \{(pair_T, pair_C)\}$
51: **end while**
52: **return** $\mathcal{D}$

---

proficiency in capturing intricate spatial structures while significantly improving semantic alignment with textual prompts compared to baseline methods. Notable improvements include enhanced

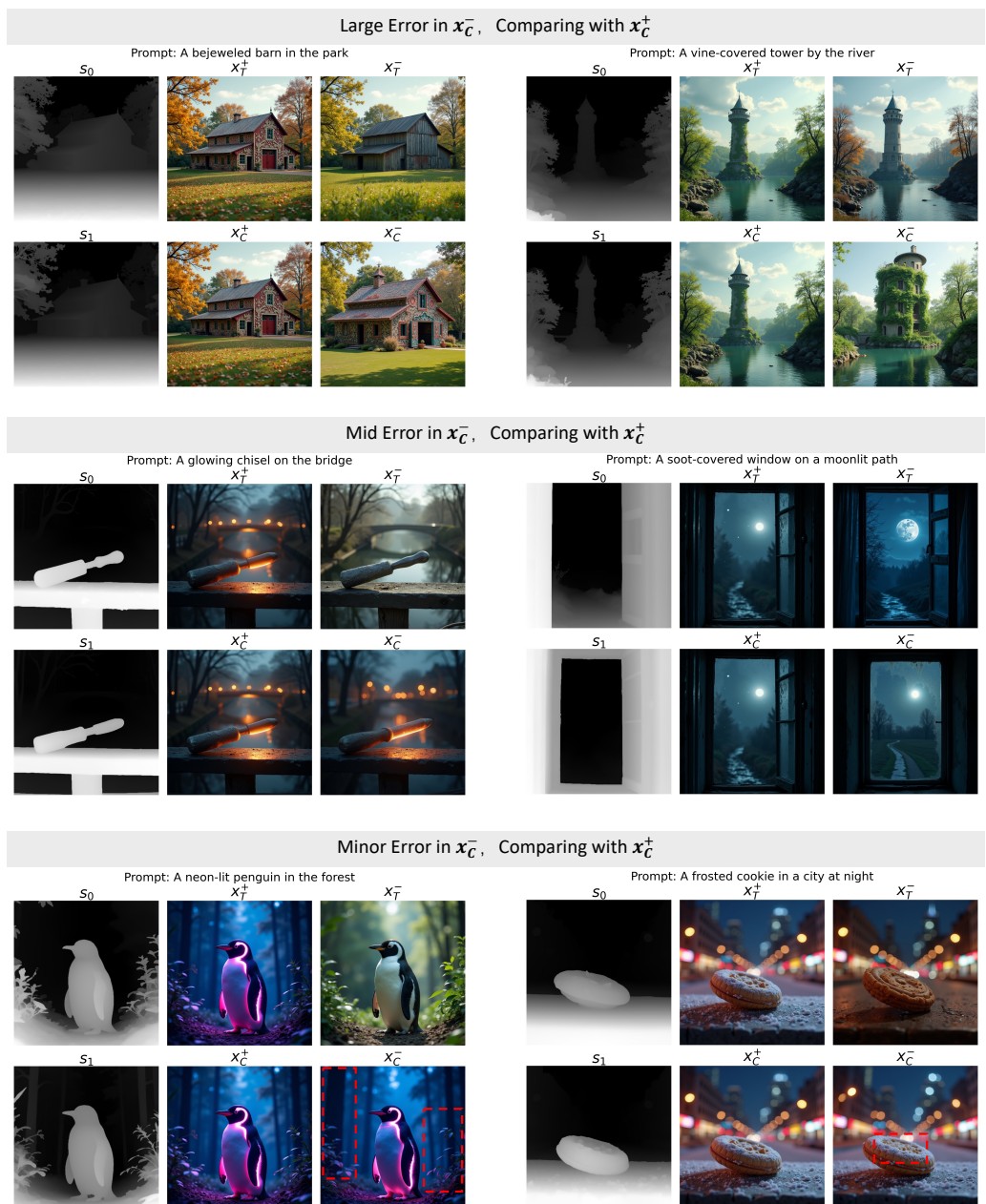

Figure 9: **Additional Examples of Disentangled and Conflict-Aware DPO Preference Data.**

textural details, such as the subtle crack patterns on candle surfaces and mossy textures on telescope bodies, demonstrating our model's ability to faithfully interpret descriptive adjectives while preserving geometric fidelity.

### F.2.2 CANNY CONDITION

Fig. 10 (Middle) showcases results from Canny edge conditions, which demand strict structural adherence. Our method excels at resolving conflicts between textual descriptions and these constraints, rectifying common failures of the baseline model. For instance, it successfully renders challenging attributes such as "scorched" or "glittering"—which the baseline struggles with—thereby significantly enhancing the control and expressive power of Canny-conditioned generation.

Figure 10: **Additional Examples of Enhancing State-of-the-Art Conditional Image Generation Methods Using Our Approach.**

### F.2.3 SOFT-EDGE CONDITION

Fig. 10 (Bottom) illustrates our approach under soft-edge conditions, which offer less stringent structural guidance while preserving overall scene composition. BideDPO maintains superior visual quality and semantic relevance even under these more ambiguous constraints, demonstrating the robustness and flexibility of our alignment framework. The results reveal enhanced textural fidelity, such as furry surfaces on binoculars and holographic effects on fountains, showcasing our model's

ability to interpret complex descriptive attributes while working within the constraints of softer structural guidance.

# G   DETAILED DERIVATION OF BIDIRECTIONALLY DECOUPLED DPO

In this section, we provide a full step-by-step derivation of our Bidirectionally Decoupled DPO (BideDPO) method, as presented in the main paper.

## G.1   FOUNDATION: INDEPENDENT LOSS COMPONENTS

Our method begins by addressing the limitation of vanilla DPO, which uses a single preference pair evaluated under a shared condition. Instead, we define two distinct preference triplets for the text and condition objectives, respectively.

For text alignment, we use the triplet $(x_T^+, x_T^-, c_0)$, where $x_T^+$ is preferred to $x_T^-$ under an initial condition $c_0$. The corresponding text loss is:

$$\mathcal{L}_{\text{text}} = -\log \sigma \left( \underbrace{f_{\text{text}}(x_T^+, c_0; \theta) - f_{\text{text}}(x_T^-, c_0; \theta)}_{\Delta_{\text{text}}(c_0; \theta)} \right) \tag{17}$$

For condition alignment, we use the triplet $(x_C^+, x_C^-, c_1)$, where $x_C^+$ is preferred to $x_C^-$ under a different, stricter condition $c_1$. The corresponding condition loss is:

$$\mathcal{L}_{\text{cond}} = -\log \sigma \left( \underbrace{f_{\text{cond}}(x_C^+, c_1; \theta) - f_{\text{cond}}(x_C^-, c_1; \theta)}_{\Delta_{\text{cond}}(c_1; \theta)} \right) \tag{18}$$

## G.2   GRADIENT DERIVATION

We now derive the partial derivative of the total loss $\mathcal{L}_{\text{decoupled}}$ with respect to the model parameters $\theta$. As noted in the main paper, the adaptive weights are computed with a stop-gradient operator, meaning they are treated as detached constants during the backward pass. Our starting point is the total loss function:

$$\mathcal{L}_{\text{decoupled}} = w_{\text{text}}\mathcal{L}_{\text{text}} + w_{\text{cond}}\mathcal{L}_{\text{cond}}. \tag{19}$$

Because $w_{\text{text}}$ and $w_{\text{cond}}$ are treated as constants with respect to $\theta$ for the gradient calculation, we can apply the sum rule directly:

$$\frac{\partial \mathcal{L}_{\text{decoupled}}}{\partial \theta} = w_{\text{text}} \frac{\partial \mathcal{L}_{\text{text}}}{\partial \theta} + w_{\text{cond}} \frac{\partial \mathcal{L}_{\text{cond}}}{\partial \theta}. \tag{20}$$

Next, we derive the gradients for the individual loss components using the chain rule. For a general loss of the form $L = -\log \sigma(z)$, its derivative is $\frac{\partial L}{\partial \theta} = -(1 - \sigma(z))\frac{\partial z}{\partial \theta}$.

Applying this to the text loss component from Eq. 17:

$$\frac{\partial \mathcal{L}_{\text{text}}}{\partial \theta} = -(1 - \sigma(\Delta_{\text{text}}(c_0; \theta))) \frac{\partial \Delta_{\text{text}}(c_0; \theta)}{\partial \theta} \tag{21}$$

And similarly for the condition loss component from Eq. 18:

$$\frac{\partial \mathcal{L}_{\text{cond}}}{\partial \theta} = -(1 - \sigma(\Delta_{\text{cond}}(c_1; \theta))) \frac{\partial \Delta_{\text{cond}}(c_1; \theta)}{\partial \theta} \tag{22}$$

Finally, substituting Eqs. 21 and 22 back into Eq. 20 gives us the final gradient for our decoupled objective, as presented in the main paper:

$$\frac{\partial \mathcal{L}_{\text{decoupled}}}{\partial \theta} = -w_{\text{text}} \left(1 - \sigma\left(\Delta_{\text{text}}(c_0; \theta)\right)\right) \frac{\partial \Delta_{\text{text}}(c_0; \theta)}{\partial \theta} - w_{\text{cond}} \left(1 - \sigma\left(\Delta_{\text{cond}}(c_1; \theta)\right)\right) \frac{\partial \Delta_{\text{cond}}(c_1; \theta)}{\partial \theta} \tag{23}$$

### G.3 Final BideDPO Objective for Diffusion Models

The general BideDPO framework can be specifically instantiated for diffusion models by defining the preference as a reward based on denoising performance. Our final objective combines the principles of decoupled losses and adaptive balancing, starting from the loss function presented in the main paper:

$$\mathcal{L}_{\text{BideDPO}}(\theta) = -\mathbb{E}_{(x_T^+, x_T^-, c_0) \sim \mathcal{D}_T, (x_C^+, x_C^-, c_1) \sim \mathcal{D}_C} \Big[ w_{\text{text}} \log \sigma(\beta T \Delta R_T) + w_{\text{cond}} \log \sigma(\beta T \Delta R_C) \Big]. \tag{24}$$

Here, $\Delta R_T$ and $\Delta R_C$ are the total reward differences for the text and condition preference pairs. And $(x_T^+, x_T^-, c_0) \sim \mathcal{D}_T, (x_C^+, x_C^-, c_1) \sim \mathcal{D}_C$ means that the samples are drawn from the text-disentangled and condition-disentangled preference pairs of the same unified preference set $\mathcal{D} = \text{zip}(\mathcal{D}_T, \mathcal{D}_C)$. We first define a per-sample reward $r(x_t, c, \epsilon; \theta)$ as the reduction in denoising error achieved by our model $\epsilon_\theta$ compared to a reference model $\epsilon_{\text{ref}}$:

$$r(x_t, c, \epsilon; \theta) = \|\epsilon - \epsilon_{\text{ref}}(x_t, c)\|^2 - \|\epsilon - \epsilon_\theta(x_t, c)\|^2. \tag{25}$$

The total reward differences are then calculated by comparing the rewards of the preferred and dispreferred samples for both the text-aligned pair (under condition $c_0$) and the condition-aligned pair (under condition $c_1$):

$$R_T = r(x_{t,T}^+, c_0, \epsilon_T^+; \theta) - r(x_{t,T}^-, c_0, \epsilon_T^-; \theta), \tag{26}$$

$$R_C = r(x_{t,C}^+, c_1, \epsilon_C^+; \theta) - r(x_{t,C}^-, c_1, \epsilon_C^-; \theta). \tag{27}$$

By substituting the definition of the reward function $r(\cdot)$ into these expressions, we can expand them to show the full formulation. For the text-aligned reward difference $R_T$:

$$R_T = \Big[ \|\epsilon_T^+ - \epsilon_{\text{ref}}(x_{t,T}^+, c_0)\|^2 - \|\epsilon_T^+ - \epsilon_\theta(x_{t,T}^+, c_0)\|^2 \Big] - \Big[ \|\epsilon_T^- - \epsilon_{\text{ref}}(x_{t,T}^-, c_0)\|^2 - \|\epsilon_T^- - \epsilon_\theta(x_{t,T}^-, c_0)\|^2 \Big]$$
$$= \|\epsilon_T^+ - \epsilon_{\text{ref}}(x_{t,T}^+, c_0)\|^2 - \|\epsilon_T^+ - \epsilon_\theta(x_{t,T}^+, c_0)\|^2 - \|\epsilon_T^- - \epsilon_{\text{ref}}(x_{t,T}^-, c_0)\|^2 + \|\epsilon_T^- - \epsilon_\theta(x_{t,T}^-, c_0)\|^2. \tag{28}$$

Similarly, for the condition-aligned reward difference $R_C$:

$$R_C = \Big[ \|\epsilon_C^+ - \epsilon_{\text{ref}}(x_{t,C}^+, c_1)\|^2 - \|\epsilon_C^+ - \epsilon_\theta(x_{t,C}^+, c_1)\|^2 \Big] - \Big[ \|\epsilon_C^- - \epsilon_{\text{ref}}(x_{t,C}^-, c_1)\|^2 - \|\epsilon_C^- - \epsilon_\theta(x_{t,C}^-, c_1)\|^2 \Big]$$
$$= \|\epsilon_C^+ - \epsilon_{\text{ref}}(x_{t,C}^+, c_1)\|^2 - \|\epsilon_C^+ - \epsilon_\theta(x_{t,C}^+, c_1)\|^2 - \|\epsilon_C^- - \epsilon_{\text{ref}}(x_{t,C}^-, c_1)\|^2 + \|\epsilon_C^- - \epsilon_\theta(x_{t,C}^-, c_1)\|^2. \tag{29}$$

Finally, substituting these fully expanded reward differences back into our main objective yields the complete BideDPO loss function used for training:

$$\mathcal{L}_{\text{BideDPO}}(\theta) = -\mathbb{E}\Big[ w_{\text{text}} \log \sigma\Big( \beta T \big[ \|\epsilon_T^+ - \epsilon_{\text{ref}}(x_{t,T}^+, c_0)\|^2 - \|\epsilon_T^+ - \epsilon_\theta(x_{t,T}^+, c_0)\|^2$$
$$- \|\epsilon_T^- - \epsilon_{\text{ref}}(x_{t,T}^-, c_0)\|^2 + \|\epsilon_T^- - \epsilon_\theta(x_{t,T}^-, c_0)\|^2 \big] \Big)$$
$$+ w_{\text{cond}} \log \sigma\Big( \beta T \big[ \|\epsilon_C^+ - \epsilon_{\text{ref}}(x_{t,C}^+, c_1)\|^2 - \|\epsilon_C^+ - \epsilon_\theta(x_{t,C}^+, c_1)\|^2$$
$$- \|\epsilon_C^- - \epsilon_{\text{ref}}(x_{t,C}^-, c_1)\|^2 + \|\epsilon_C^- - \epsilon_\theta(x_{t,C}^-, c_1)\|^2 \big] \Big) \Big]. \tag{30}$$

## H Additional Discussions

### H.1 Discussion on More DPO Variants

In our main experiments, we compare BideDPO against a "naive" application of DPO, as described in the Baseline Configurations section. This Naive DPO baseline uses a single preference pair format: the positive sample aligns well with both text and condition $(T^+, C^+)$, while the negative sample aligns poorly with both $(T^-, C^-)$. This setup, while straightforward, does not fully capture the complexity of the alignment problem, where conflicts can arise from either text or condition independently.

To provide a more robust comparison, we introduce an additional baseline, "DPO (Mixed)". In this setting, the negative samples are constructed from a mix of failure cases: poor text and poor condition $(T^-, C^-)$, good text but poor condition $(T^+, C^-)$, and poor text but good condition $(T^-, C^+)$. This creates a more diverse and challenging training signal for the DPO model, forcing it to learn a more nuanced reward function.

The results, presented in Tab. 9, reveal an interesting trade-off. The DPO (Mixed) baseline achieves a higher Success Ratio (0.73 vs. 0.71) and CLIP score (0.2884 vs. 0.2860) compared to the Naive DPO, indicating improved text alignment. However, it performs worse on conditional fidelity, with higher (worse) MSE and SGMSE scores. This suggests that while a mixed-negative strategy helps the model better understand textual nuances, the undifferentiated DPO loss struggles to balance the competing objectives, leading to a degradation in structural adherence.

In contrast, our BideDPO method significantly outperforms both DPO baselines across all metrics. By explicitly decoupling the preference pairs for text and condition alignment, BideDPO provides clear, unambiguous learning signals for each objective. This, combined with our adaptive weighting mechanism, allows the model to simultaneously improve both text-prompt consistency and conditional fidelity, overcoming the trade-offs that limit standard DPO approaches.

| Method | SR ↑ | MSE ↓ | SGMSE ↓ | CLIP ↑ |
|---|---|---|---|---|
| Union-Pro2 | 0.49 | 176.982 | 272.400 | 0.2748 |
| + DPO (Naive) | 0.71 | 168.284 | 219.935 | 0.2860 |
| + DPO (Mixed) | 0.73 | 186.417 | 229.857 | 0.2884 |
| + Ours | **0.84** | **163.968** | **195.728** | **0.2924** |

Table 9: Comparison of different DPO configurations for depth-conditioned image generation. Our method surpasses both naive and mixed DPO baselines.

## H.2 COMPARISON WITH MORE IMAGE CONDITIONAL GENERATION METHODS

To evaluate BideDPO against state-of-the-art approaches for conditional image generation, we compare with LooseControl (Bhat et al., 2024) and ControlNet++ (Li et al., 2024).

### H.2.1 RESULTS ON DUALALIGN BENCHMARK

Tab. 10 presents the quantitative comparison on the DualAlign benchmark with depth conditioning. BideDPO achieves significantly superior performance across all metrics, with a success rate (SR) of **0.84**, MSE of **164.0**, SGMSE of **195.7**, and CLIP score of **0.2924**. In contrast, LooseControl and ControlNet++ show limited performance, with SR values of 0.43 and 0.49, respectively, and significantly higher structural errors (MSE: 791.13 and 331.85).

### H.2.2 RESULTS ON COCO BENCHMARK

Tab. 11 shows the comparison on the COCO benchmark with depth conditioning. BideDPO again achieves superior performance, with an SR of **0.91**, MSE of **236.3**, SGMSE of **245.3**, and CLIP score of **0.2633**. LooseControl and ControlNet++ show lower success rates (0.72 and 0.79) and significantly higher structural errors.

### H.2.3 DISCUSSION

The comparison reveals that existing approaches alone are insufficient to resolve conflicts between text and condition constraints. In contrast, BideDPO's post-training approach with bidirectionally decoupled objectives enables it to effectively harmonize both constraints without sacrificing either objective.

## H.3 COMPARISON WITH DPO-BASED POST-TRAINING METHODS

To evaluate BideDPO against state-of-the-art DPO-based post-training methods for conditional image generation, we compare with SPO (Liang et al., 2024) and RankDPO (Karthik et al., 2024). These methods are based on DPO with some improvements for preference optimization, but differ

Table 10: **Comparison with more image conditional generation methods on DualAlign benchmark.**

| Method | SR ↑ | MSE ↓ | SGMSE ↓ | CLIP ↑ |
|---|---|---|---|---|
| LooseControl | 0.43 | 791.13 | 1280.17 | 0.2852 |
| ControlNet++ | 0.49 | 331.85 | 480.66 | 0.2854 |
| **BideDPO (ours)** | **0.84** | **164.0** | **195.7** | **0.2924** |

Table 11: **Comparison with more image conditional generation methods on COCO benchmark.**

| Method | SR ↑ | MSE ↓ | SGMSE ↓ | CLIP ↑ |
|---|---|---|---|---|
| LooseControl | 0.72 | 1334.01 | 1705.97 | 0.2534 |
| ControlNet++ | 0.79 | 548.26 | 668.92 | 0.2557 |
| **BideDPO (ours)** | **0.91** | **236.3** | **245.3** | **0.2633** |

from our approach in how they handle the dual objectives of text alignment and conditional fidelity. Note that RankDPO is not publicly available, so we implement it based on the methodology described in the paper. All methods share the same FLUX backbone and evaluation pipeline to ensure fair comparison.

### H.3.1 RESULTS ON DUALALIGN BENCHMARK

Tab. 12 presents the quantitative comparison on the DualAlign benchmark with depth conditioning. BideDPO achieves the best performance across all metrics, with a success rate (SR) of **0.84**, MSE of **164.0**, SGMSE of **195.7**, and CLIP score of **0.2924**.

SPO achieves a competitive SR of 0.78 with good structural control (MSE: 166.2, SGMSE: 208.7), while RankDPO reaches a SR of 0.83 but suffers from higher structural errors (MSE: 188.5, SGMSE: 235.6). Both methods struggle to fully resolve conflicts between text and condition constraints.

In contrast, BideDPO's bidirectionally decoupled objective effectively balances both text alignment and structural conditioning by explicitly separating preference pairs along text and condition axes, enabling the model to simultaneously improve both text-prompt consistency and conditional fidelity.

### H.3.2 DISCUSSION

The comparison with DPO-based methods reveals different trade-offs: SPO maintains better structural control but achieves lower text alignment, while RankDPO improves text alignment but struggles with structural fidelity. This suggests that existing DPO-based methods, while effective, do not fully address the challenge of simultaneously optimizing both text and condition objectives when they conflict.

BideDPO's explicit decoupling of text and condition objectives, combined with adaptive loss balancing, enables it to outperform both baselines by effectively harmonizing both constraints without sacrificing either objective. The bidirectionally decoupled approach provides clearer learning signals for each objective, allowing the model to learn more effectively from preference pairs that may be ambiguous when both objectives are considered together.

### H.4 STABLE DIFFUSION 1.5 + BIDEDPO

**Stable Diffusion 1.5 + BideDPO.** To demonstrate that BideDPO is not limited to FLUX-based models but also generalizes to Stable Diffusion-based architectures, we further fine-tune the ControlNet of Stable Diffusion 1.5 with our bidirectionally decoupled objective. To highlight how BideDPO improves controllable generation under the DualAlign depth benchmark, we compare our approach with the ControlNet baseline. Relative to ControlNet, our model dramatically increases success rate while simultaneously lowering both MSE and SGMSE, indicating superior adherence to depth conditioning without sacrificing semantic fidelity. Notably, the CLIP score also rises, underscoring that the additional control signal does not compromise text alignment. The detailed quantitative comparison is provided in Tab. 13, while Fig. 11 visualizes the accompanying qualitative gains.

Table 12: **Comparison with DPO-based post-training methods on DualAlign benchmark.** All methods share the same FLUX backbone and evaluation pipeline.

| Method | SR ↑ | MSE ↓ | SGMSE ↓ | CLIP ↑ |
|---|---|---|---|---|
| SPO | 0.78 | 166.2 | 208.7 | 0.2881 |
| RankDPO | 0.83 | 188.5 | 235.6 | 0.2914 |
| **BideDPO (Ours)** | **0.84** | **164.0** | **195.7** | **0.2924** |

Table 13: **Stable Diffusion 1.5 depth-conditioned image generation on DualAlign Benchmark.** "Ctrl." indicates support for conditional generation; qualitative comparisons appear in Fig. 11.

| Method | Ctrl. | SR ↑ | MSE ↓ | SGMSE ↓ | CLIP ↑ |
|---|---|---|---|---|---|
| SD 1.5-ControlNet | ✓ | 0.50 | 391.80 | 592.92 | 0.2824 |
| + BideDPO (Ours) | ✓ | **0.71** | **187.44** | **234.14** | **0.2853** |

## Stable Diffusion Depth

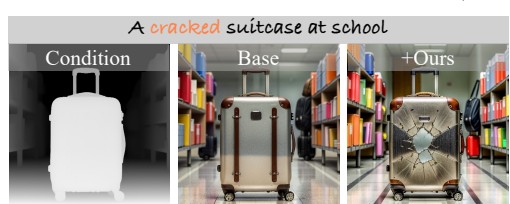
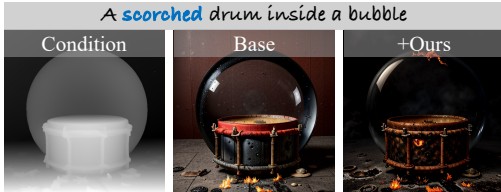

Figure 11: **Stable Diffusion 1.5 depth-conditioned generation on DualAlign.** Qualitative comparison between the ControlNet baseline and our BideDPO fine-tuned model. BideDPO preserves the textual semantics while aligning more faithfully with the provided depth controls, yielding sharper geometry and cleaner spatial layouts.

### H.5    MULTI-CONDITION GENERATION: TEXT + DEPTH + CANNY

**Scaling to Multiple Simultaneous Conditions.** To demonstrate that BideDPO scales beyond two conditioning modalities, we showcase a challenging multi-condition generation scenario that simultaneously enforces text prompts, depth maps, and Canny edge constraints. As illustrated in Fig. 12, we condition the generation on both depth and Canny edge maps extracted from an original anime figurine image, while applying the text prompt "A jade-like Anime figurine." This setup creates a complex multi-objective optimization problem where the model must harmonize three distinct constraints: (1) textual semantics (jade-like material transformation), (2) depth geometry (preserving 3D spatial structure), and (3) edge structure (maintaining fine-grained boundaries and details).

Our BideDPO method successfully balances all three objectives, producing a high-quality jade-like figurine that preserves both the global spatial layout from the depth map and the fine-grained details captured by the Canny edges, while faithfully realizing the translucent, polished jade aesthetic described in the text prompt. In contrast, UnionPro2 struggles to simultaneously satisfy all constraints: with low conditioning strength, it loses structural fidelity to the depth and edge maps; with default conditioning strength, it maintains better structural alignment but fails to fully realize the style transformation, resulting in a less convincing jade-like appearance. This example demonstrates that BideDPO's decoupled objective and adaptive loss balancing mechanism naturally extend to handle multiple conditioning inputs by treating each conditioning path independently and dynamically adjusting their relative importance during training.

Tab. 14 provides quantitative evidence that BideDPO effectively handles simultaneous depth and Canny edge conditioning. When evaluated on the DualAlign depth benchmark, our method achieves a success rate (SR) of 0.81, significantly outperforming Union-Pro2's 0.49, while simultaneously reducing both MSE (159.6 vs. 177.0) and SGMSE (197.5 vs. 272.4) errors, indicating superior structural fidelity. The CLIP score also improves from 0.2748 to 0.2901, demonstrating enhanced text alignment. On the Canny benchmark, BideDPO maintains strong performance with SR of 0.68 (vs. 0.34 for Union-Pro2), F1 score of 0.594 (vs. 0.418), and SG F1 of 0.381 (vs. 0.143), while improving CLIP from 0.2753 to 0.2857. These results confirm that BideDPO's bidirectional decou-

Table 14: **Results for multi-condition generation (depth + canny) on DualAlign Benchmark.**
The model is simultaneously conditioned on both depth and canny edge maps. Left half shows
depth-conditioned results, right half shows canny-conditioned results.

| | Depth Benchmark | | | | Canny Benchmark | | | |
|---|---|---|---|---|---|---|---|---|
| Method | SR ↑ | MSE ↓ | SGMSE ↓ | CLIP ↑ | SR ↑ | F1 ↑ | SG F1 ↑ | CLIP ↑ |
| Union-Pro2 | 0.49 | 177.0 | 272.4 | 0.2748 | 0.34 | 0.418 | 0.143 | 0.2753 |
| + Ours (depth+canny merge) | **0.81** | **159.6** | **197.5** | **0.2901** | **0.68** | **0.594** | **0.381** | **0.2857** |

pling mechanism successfully harmonizes multiple conditioning modalities without compromising
performance on either benchmark, validating the method's scalability to complex multi-condition
generation scenarios.

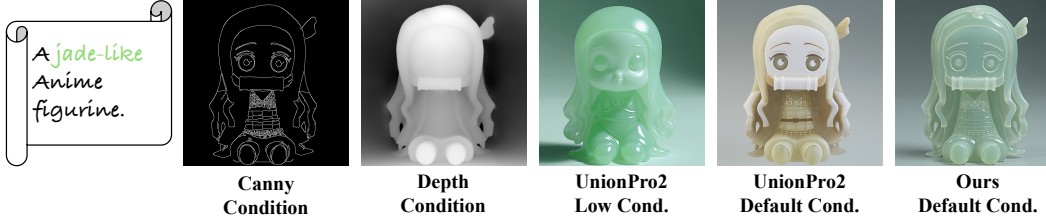

**Canny Condition**     **Depth Condition**     **UnionPro2 Low Cond.**     **UnionPro2 Default Cond.**     **Ours Default Cond.**

Figure 12: **Multi-condition generation with text, depth, and Canny edge controls.** Example
demonstrating BideDPO's capability to handle multiple simultaneous conditioning inputs. The text
prompt is "A jade-like Anime figurine." We condition the generation on both depth maps and Canny
edges extracted from the original image, while applying the jade-like style transformation. Our
method (rightmost) successfully harmonizes all three constraints—textual semantics, depth geome-
try, and edge structure—producing a high-fidelity jade-like figurine that preserves both spatial layout
and fine-grained details. In contrast, UnionPro2 struggles to balance these competing objectives, ei-
ther losing structural fidelity (low cond) or failing to fully realize the style transformation (default
cond).

## H.6 DISCUSSION ON ADAPTIVE LOSS BALANCING (ALB) METHODS

We investigate the robustness of ALB to different weight calculation methods and batch sizes.

### H.6.1 INSTANCE MEAN VS. HISTORICAL MEAN

We compare our default instance mean approach with a historical mean (moving average) approach.
As shown in Tab. 15, both methods achieve similar performance (SR: **0.84**), with minimal differ-
ences across metrics. This robustness stems from our normalization scheme that prevents unstable
weight fluctuations, making the simpler instance mean approach a practical choice.

### H.6.2 BATCH SIZE SENSITIVITY

We evaluate ALB with batch sizes of 8, 16, and 32. Tab. 15 shows consistent performance across all
batch sizes (SR: 0.83-0.85), demonstrating that ALB maintains effective loss balancing regardless
of batch size.

Table 15: Comparison of different ALB methods and batch size sensitivity.

| Method | SR ↑ | MSE ↓ | SGMSE ↓ | CLIP ↑ |
|---|---|---|---|---|
| ALB instance mean (batch=8) | **0.84** | **164.0** | **195.7** | 0.2924 |
| ALB instance mean (batch=16) | 0.83 | 167.4 | 199.2 | **0.2940** |
| ALB instance mean (batch=32) | **0.85** | 160.7 | 195.5 | 0.2900 |
| ALB historical mean | 0.84 | 166.2 | 198.8 | **0.2934** |

### H.7 DISCUSSION ON THE NUMBER OF ITERATIONS

As shown in the ablation study in the main paper (Tab. 5), the performance of our method improves steadily from the baseline up to Iteration 3, which we selected as our final model. However, we observed a slight degradation in performance at Iteration 4. This phenomenon suggests a potential for overfitting and highlights the trade-offs inherent in our iterative optimization strategy.

We hypothesize that this performance drop is due to the model beginning to overfit to the biases of our automated data generation and scoring pipeline. While the iterative process is highly effective at bootstrapping performance, it creates a feedback loop where the model is trained exclusively on data it generates itself. After several iterations, the data distribution, while high-quality, may become narrower and reflect the specific quirks of the generator and the VLM used for scoring.

At Iteration 4, the model may start to fit to these artifacts rather than learning a more generalizable representation of text and condition alignment. The preference pairs generated may also become less informative, as the distinction between "preferred" and "dispreferred" samples becomes increasingly subtle for an already powerful generator. Iteration 3 appears to represent the optimal balance point, where the model has reaped the benefits of high-quality, self-generated data without yet succumbing to the effects of overfitting to its own narrowing data distribution.

### H.8 DETAILED DISCUSSION ON VLM SELECTION

In our evaluation pipeline, we employ Qwen2.5-VL-72B as the primary VLM for assessing text-image alignment through the SR metric. This choice warrants careful justification, as the reliability of our conclusions depends on the quality and consistency of the evaluator.

**Rationale for Qwen2.5-VL-72B.** We deliberately adopt Qwen2.5-VL-72B as our primary judge because it is fully open-source and freely usable, and has become a de-facto standard VLM in recent academic work on text–image evaluation. This makes our pipeline easier to reproduce and our SR metric easier to compare against future papers—even if Qwen is not always the single most SOTA model on every benchmark. In addition, Qwen offers strong coverage across diverse object categories and scene types, which is important for our broad DualAlign setting. Its accessibility and community acceptance make it a practical, "standard" evaluator that lowers the barrier for future work to build on our method.

**Cross-VLM and Human Validation.** To ensure that our conclusions are not artifacts of a specific VLM, we re-evaluate the same test set with GPT-4o and human raters. The results demonstrate strong consistency across all evaluators: As shown in Tab. 8, all three evaluators consistently rank $+Ours > +DPO > +SFT > UnionPro2$, with BideDPO achieving the highest scores across all judges (0.82–0.84). Notably, Qwen and Human evaluators both assign **0.84** to our method, demonstrating that Qwen serves as a reliable proxy for human judgment. The consistent relative rankings across different evaluators validate that our conclusions are robust and not artifacts of a specific VLM, confirming that using Qwen as a representative VLM evaluator is appropriate.