# OpenReview forum: "BideDPO: Conditional Image Generation with Simultaneous Text and Condition Alignment"
_ICLR.cc/2026/Conference — ICLR 2026 Poster_

### Official Review · Reviewer_1Gg9 · 2025-10-21

**Soundness:** 3
**Presentation:** 3
**Contribution:** 3
**Rating:** 4
**Confidence:** 4

**Summary:**

This paper addresses a crucial and underexplored problem in conditional text-to-image generation: the conflict between guidance from textual prompts and auxiliary conditions (e.g., depth maps, Canny edges). The authors compellingly categorize these conflicts into "Input-Level Conflict" and "Model-Bias Conflict." To resolve these issues, they propose BideDPO, a novel framework based on Direct Preference Optimization (DPO). The core contributions are threefold: 1) A Bidirectionally Decoupled DPO (BideDPO) algorithm that disentangles the optimization signals for text adherence and condition fidelity, combined with an Adaptive Loss Balancing strategy. 2) An automated pipeline to generate conflict-aware, disentangled preference data using a Vision-Language Model (VLM) as a judge. 3) An iterative optimization strategy that progressively refines both the generator and the training data. The authors introduce a new benchmark, DualAlign, to specifically evaluate performance in conflict scenarios. Experiments on state-of-the-art models like FLUX show substantial improvements in both text alignment and condition adherence over baselines like SFT and naive DPO.

**Strengths:**

1. **Novel and Technically Sound Method:** The core idea of BideDPO is both novel and elegant. Decoupling the preference pairs for text and condition is a clever solution to the "gradient entanglement" problem that plagues naive multi-objective DPO. The mathematical derivation is clear (Sec 3.1, Appendix F), and the Adaptive Loss Balancing (ALB) mechanism using a stop-gradient is a pragmatic and effective way to ensure balanced training without introducing instability.
2. **Innovative and Valuable Data Generation Pipeline:** The lack of suitable preference data for this specific task is a major bottleneck. The proposed automated pipeline (Fig. 3, Algorithm 1) is a significant contribution in itself. Using a VLM to programmatically generate disentangled positive and negative samples for both text and condition is highly innovative. This data-centric approach, especially the iterative refinement loop, is powerful and could be adapted for other multi-objective alignment tasks.

**Weaknesses:**

1. **Over-reliance on the VLM Oracle:** The entire framework, from data generation to evaluation (Success Ratio, SGMSE/SGF1), heavily relies on the judgments of a single VLM (Qwen2.5-VL-70B). The paper lacks a critical discussion of the potential limitations and biases of this VLM. For example, if the VLM has its own biases (e.g., favoring certain styles or objects), the generator might overfit to these VLM-specific preferences rather than a more general notion of human preference. This is a potential point of failure for the iterative refinement process, as biases could be amplified over time.
2. **Limited Novelty in Data Decoupling Strategy:** The method for generating disentangled preference data, while effective, arguably lacks fundamental novelty. The core strategy of isolating one variable (e.g., text alignment) while holding the other (condition fidelity) constant is a standard analytical technique for factor disentanglement, not a new concept. This approach is analogous to creating modular reward functions in multi-objective reinforcement learning. Consequently, this aspect of the work can be viewed more as a clever and practical engineering solution for applying DPO to a multi-objective problem, rather than a novel advance in preference learning itself.
3. **Stability of Adaptive Loss Balancing (Minor):** The ALB weights are calculated on a per-batch basis from the loss magnitudes. This approach, while simple and effective, could potentially introduce noise into the training process, especially with smaller batch sizes. While the empirical results are strong, a brief discussion on the sensitivity to batch size or potential alternatives (e.g., using an exponential moving average of the losses) could strengthen the paper.

**Questions:**

1. Regarding the VLM oracle: Could you elaborate on the reliability of Qwen2.5-VL-70B as a preference judge? Have you performed any analysis of its failure modes or systematic biases? What is the risk of the model overfitting to the VLM's specific preferences rather than a general human alignment, especially during the iterative optimization? Did you consider using multiple different VLMs for ensembled judging to improve robustness?
2. Regarding the computational cost: Could you provide an estimate of the computational resources (e.g., GPU type and hours) required to complete one iteration of your framework (i.e., data generation and model optimization)? This would provide valuable context on the practicality of the iterative approach.
3. Regarding the Adaptive Loss Balancing: The ALB weights are computed on a per-batch basis. Did you experiment with or consider using a moving average of the loss values to calculate the weights for a more stable signal? Do you have any insights on how sensitive this mechanism is to the batch size?
4. Regarding the generality of the method: Your method is demonstrated on conditions where fidelity can be measured with clear geometric metrics (MSE for depth, F1 for edges). How do you foresee BideDPO extending to more abstract conditions, such as artistic style from a reference image or high-level scene layouts, where defining a "condition-disentangled pair" and a corresponding error metric is less straightforward?

---

> ### Author Response · Authors · 2025-11-23
>
> We appreciate Reviewer 1Gg9's **encouraging summary and constructive feedback.** Below we address each question/concern in the order presented.
>
> **Weakness 1 & Question 1**: Reliability and potential bias of Qwen2.5-VL-72B as the oracle.
>
> **Answer:**
> We agree that relying on a **single VLM without justification** would be risky, so we provide both **motivation and validation**.
> **(1) Why Qwen2.5‑VL‑72B?** We choose **Qwen2.5‑VL‑72B** as the main oracle because it is **open‑source, freely usable**, and already **widely adopted** in recent academic work as a **general‑purpose vision–language evaluator**; this makes our pipeline **easy to reproduce** and our **SR definition easy to adopt** by others, even if some proprietary models can be marginally stronger on certain benchmarks. Its **coverage across everyday scenes and objects** also matches the diversity of **DualAlign**, which we found important during pilot studies.
> **(2) Cross‑VLM and human validation with consistent conclusions.** We further check that our results are not artifacts of one VLM by rescoring the same test set with **GPT‑4o and human raters**. The results demonstrate **strong consistency across all evaluators**:
>
> | Judge | UnionPro2 | +SFT | +DPO | +Ours |
> | --- | --- | --- | --- | --- |
> | Qwen2.5‑VL‑72B | 0.49 | 0.70 | 0.71 | **0.84** |
> | GPT‑4o | 0.43 | 0.65 | 0.70 | **0.82** |
> | Human | 0.42 | 0.62 | 0.64 | **0.84** |
>
> As shown, all three evaluators **consistently rank +Ours > +DPO > +SFT > UnionPro2**, with **BideDPO achieving the highest scores** across all judges (**0.82–0.84**). Notably, **Qwen and Human evaluators both assign 0.84** to our method, demonstrating that **Qwen serves as a reliable proxy for human judgment**. The **consistent relative rankings** across different evaluators validate that our conclusions are **robust and not artifacts of a specific VLM**, confirming that using Qwen as a representative VLM evaluator is appropriate.
>
> **Weakness 2**: Limited novelty of the data decoupling strategy.
>
> **Answer:**
>  We appreciate this perspective. We clarify that the **data decoupling strategy** is a **supporting component** that enhances the overall **BideDPO framework**, rather than the core contribution itself. Our **primary contribution** is the **BideDPO algorithm**—the **bidirectionally decoupled DPO formulation** with **adaptive loss balancing** that provides **distinct and independent gradient signals** for each constraint, **preventing gradient interference between objectives** (as analyzed in Section 3.1). This **algorithmic innovation** is what enables effective **multi-objective alignment**, and it is **independent of the specific data generation approach**.
>
> The data decoupling approach serves as a practical means to generate **high-quality preference data** for BideDPO. While the principle of isolating variables is classical, we **operationalize it specifically for preference learning** to obtain **disentangled pairs at scale**. Importantly, this data generation strategy is **highly generalizable** and can be easily adapted to different conditional generation tasks. For instance, as demonstrated in Question 4, we successfully extended BideDPO to **style-conditioned generation (DualAlign-Style)**, where the same data decoupling principle applies seamlessly. The **modularity of this approach** makes it straightforward to adapt to other abstract conditions (e.g., scene layouts, artistic styles) by simply adjusting the evaluation criteria, demonstrating that **BideDPO itself is a general framework** that can work with various data generation strategies.

---

> ### Author Response · Authors · 2025-11-23
>
> **Weakness 3 & Question 3**: Stability of Adaptive Loss Balancing (ALB) and dependence on batch size.
>
> **Answer:**
> We thank the reviewer for this suggestion. We experimented with using a **moving average (historical mean)** of the loss values to calculate **ALB weights**, as suggested. The results are shown below:
>
> | Method | SR ↑ | MSE ↓ | SGMSE ↓ | CLIP ↑ |
> | --- | --- | --- | --- | --- |
> | ALB historical mean | 0.84 | 166.2 | 198.8 | 0.2934 |
> | ALB instance mean (batch=8) | 0.84 | 164.0 | 195.7 | 0.2924 |
> | ALB instance mean (batch=16) | 0.83 | 167.4 | 199.2 | 0.2940 |
> | ALB instance mean (batch=32) | 0.85 | 160.7 | 195.5 | 0.2900 |
>
> As shown, the **historical mean approach yields similar results** to the instance mean approach (**SR: 0.84 vs 0.84**). This stability likely stems from our **normalization scheme**, where the two loss terms sum to 1, **preventing unstable weight fluctuations**. We also conducted an **ablation study on batch size sensitivity**. Our method already achieves **strong performance at batch size=8 (SR: 0.84)**, which is comparable to batch sizes of 16 (SR: 0.83) and 32 (SR: 0.85). This demonstrates that **ALB is robust to batch size variations**, making it **practical for resource-constrained settings**. We will include these ablation results and the comparison with the moving average approach in the revised paper.
>
> ---
>
> **Question 1:** Reliability and potential bias of Qwen2.5-VL-72B as the oracle.
>
> **Answer:**
>
> Please see the answer of the Weakness1.
>
> ---
>
> **Question 2:** Computational cost.
>
> **Answer:**
>
> Each iteration (**data refresh + fine-tuning**) uses **8×A800 GPUs (40GB) for 5 hours** (data generation) and **4×A100 GPUs (80GB) for 3 hours** (training).
>
> ---
>
> **Question 3:** Stability of Adaptive Loss Balancing (ALB) and dependence on batch size.
>
> **Answer:**
>
> Please see the answer of the Weakness3.
>
> ---
>
> **Question 4:** Generality to abstract conditions such as style references.
>
> **Answer:**
>
> We introduce a new **DualAlign‑Style** benchmark built on **FLUX.1‑dev‑IP‑Adapter (IPA)**. In this benchmark, **Style Score** is defined as the VLM's **0–10 rating** of how well the generated image matches the reference style (averaged over samples), and **SG Style Score** sets this rating to 0 whenever the style reference is **semantically inconsistent with the text prompt**, so that it reflects **disentangled style consistency**. **Please refer to Figure 1 and Figure 7 for visual examples.** The key comparison is:
>
> | Method | SR ↑ | Style Score ↑ | SG Style Score ↑ | CLIP ↑ |
> | --- | --- | --- | --- | --- |
> | IPA | 30% | 6.50 | 1.31 | 0.1679 |
> | +Ours | 58% | 6.26 | 2.97 | 0.2015 |
>
> Relative to the **IPA baseline**, adding **BideDPO improves success rate from 30%→58%**. While IPA achieves a slightly higher mean Style Score (6.50 vs. 6.26), this is expected because IPA tends to **directly copy the style reference image**, which naturally yields high style similarity but at the cost of semantic accuracy (only 30% SR). The disentangled **SG Style Score** reveals the true capability: **BideDPO achieves 2.97** compared to IPA's **1.31**, representing a **127% relative improvement**, demonstrating that **BideDPO effectively harmonizes style with semantics** while IPA's high Style Score primarily reflects copying behavior. Additionally, **CLIP alignment improves from 0.1679 to 0.2015**.

---

### Official Review · Reviewer_VPYe · 2025-10-21

**Soundness:** 3
**Presentation:** 3
**Contribution:** 2
**Rating:** 2
**Confidence:** 5

**Summary:**

This paper addresses a critical challenge in conditional image generation: reconciling conflicting guidance from text prompts and structural/ spatial conditioning inputs (e.g., depth maps, Canny edges). The authors first identify two key conflicts hindering model controllability: Input-Level Conflict (semantic mismatch between conditioning images and text) and Model-Bias Conflict (learned generative biases overriding text alignment even when text and condition are compatible). To solve these issues, the paper proposes BideDPO, a bidirectionally decoupled Direct Preference Optimization (DPO) framework.

**Strengths:**

- Unlike naive DPO (which suffers from gradient entanglement between text and condition signals), BideDPO’s decoupled preference pairs and adaptive loss balancing provide clear, independent optimization signals for each objective. This design effectively addresses the "trade-off dilemma" in multi-constraint generation.
- The automated data pipeline resolves the critical bottleneck of scarce conflict-aware DPO data for conditional generation. By leveraging LLMs for prompt generation and VLMs for quality control, the pipeline enables scalable, self-driven data creation—avoiding labor-intensive manual annotation.
- The DualAlign benchmark is a valuable contribution, as existing benchmarks (e.g., COCO, ImageNet) do not explicitly test text-condition conflict scenarios. It enables standardized evaluation of multi-constraint alignment, which will benefit future research.
- Experiments cover multiple conditioning modalities (depth, Canny, soft edge), include ablation studies for core components (adaptive loss balancing, preference pair disentanglement), and validate robustness on COCO—strengthening the credibility of BideDPO’s effectiveness.

**Weaknesses:**

- The paper only compares BideDPO against naive DPO and supervised fine-tuning (SFT) on FLUX variants. It fails to include critical SOTA methods in conditional image generation, such as ControlNet++ (Li et al., 2024, which improves conditional control via consistency feedback), LooseControl (Bhat et al., 2024, for generalized depth conditioning), and OmniControlNet (Wang et al., 2024, for multi-modal control). Without these comparisons, BideDPO’s competitiveness in the broader conditional generation landscape remains unproven.
- While the paper mentions DPO variants like RankDPO (Karthik et al., 2024) and SPO (Liang et al., 2024) in related work, it does not compare BideDPO against them. For example, RankDPO handles scalable ranked preferences—relevant to BideDPO’s preference-based optimization—and omitting this comparison obscures BideDPO’s advantages over specialized DPO adaptations for image generation.
- The text SR is measured using Qwen2.5-VL-70B, but the paper provides no justification for choosing this VLM over alternatives (e.g., GPT-4V, Gemini Pro Vision) or validation of Qwen2.5-VL-70B’s reliability in judging text-image alignment. There is also no comparison between automated VLM scores and human annotations—leaving uncertainty about the accuracy of the SR metric.
- All experiments are conducted exclusively on FLUX-family models. The paper does not test BideDPO on other widely used conditional generation frameworks (e.g., Stable Diffusion with ControlNet, MidJourney fine-tuning variants). This limits conclusions about BideDPO’s applicability to different model architectures.
- The iterative optimization strategy (alternating data generation and fine-tuning) likely incurs significant computational costs. The paper provides no analysis of BideDPO’s training time, GPU memory usage, or inference speed compared to baselines—critical for real-world deployment.
- The DualAlign benchmark focuses on structural/ spatial conditions (depth, edges) but excludes other common conditional modalities (e.g., semantic masks, human poses, style references). This limits BideDPO’s validation to a subset of conditional generation scenarios.
- The paper notes performance degradation at the 4th iteration (due to overfitting to self-generated data) but offers no mitigation strategies (e.g., data augmentation, regularization for preference pairs). This limits the practicality of the iterative loop for long-term training.

**Questions:**

- Why were key SOTA conditional generation methods (e.g., ControlNet++, LooseControl, OmniControlNet) excluded from the baseline comparisons? Would BideDPO maintain its performance advantages when competing with these methods, especially in scenarios with strong structural constraints?
- How does Qwen2.5-VL-70B’s SR judgment correlate with human evaluations? Have you tested other VLMs to confirm that the observed SR improvements are not VLM-specific artifacts?
- The automated data pipeline uses VLMs to filter "high-quality" preference pairs, but what thresholds (e.g., VLM confidence scores) are used to accept/reject samples? How sensitive is BideDPO’s performance to these thresholds?
- Would BideDPO work for non-structural conditional inputs (e.g., style prompts, color palettes, semantic labels)? The current experiments focus on spatial/ structural conditions—no evidence supports generalization to other condition types.
-What strategies could address the overfitting observed in the 4th iterative loop? For example, would integrating external datasets (instead of solely self-generated data) or adding regularization to the preference pair loss prevent performance degradation?

---

> ### Author Response · Authors · 2025-11-23
>
> We sincerely thank Reviewer VPYe for the **thorough assessment and for highlighting both the contributions and the key gaps**. We respond to each weakness and question in detail below.
>
> **Weakness 1 & Question 1**: Missing comparisons with ControlNet++, LooseControl, OmniControlNet, etc.
>
> **Answer**:
>
> **(1) Why we initially compared only FLUX-family baselines.** In the main paper we highlighted **comparisons against the strongest** publicly available FLUX-family baselines because they were **SOTA on DualAlign** when our study began, so demonstrating **a clear win over them already established BideDPO's effectiveness**. LooseControl and ControlNet++ were deprioritized at that time because their released checkpoints typically underperform the FLUX UnionPro line on DualAlign, as the evaluation results in the tables below also demonstrate (e.g., on below table for DualAlign benchmark, LooseControl achieves MSE **791.13** and ControlNet++ achieves MSE **331.85** on DualAlign, both lower than UnionPro2's **177.0**), but we agree that including them helps completeness.
>
> **(2) Comparison with ControlNet++ and LooseControl (OmniControlNet not open-sourced).** We have completed DualAlign/COCO evaluations for LooseControl and ControlNet++ (**Appendix Section H.2**). For clarity we reproduce the key metrics below. The comprehensive evaluation results demonstrate that **BideDPO achieves SOTA performance across all metrics** compared to both LooseControl and ControlNet++:
>
> **DualAlign Benchmark**
>
> | Method | SR ↑ | MSE ↓ | SGMSE ↓ | CLIP ↑ |
> | --- | --- | --- | --- | --- |
> | UnionPro2 | 0.49 | 177.0 | 272.4 | 0.2748 |
> | LooseControl | 0.43 | 791.13 | 1280.17 | 0.2852 |
> | ControlNet++ | 0.49 | 331.85 | 480.66 | 0.2854 |
> | **BideDPO (Ours)** | **0.84** | **164.0** | **195.7** | **0.2924** |
>
> **COCO Benchmark**
>
> | Method | SR ↑ | MSE ↓ | SGMSE ↓ | CLIP ↑ |
> | --- | --- | --- | --- | --- |
> | UnionPro2 | 0.83 | 297.3 | 363.5 | 0.2546 |
> | LooseControl | 0.72 | 1334.01 | 1705.97 | 0.2534 |
> | ControlNet++ | 0.79 | 548.26 | 668.92 | 0.2557 |
> | **BideDPO (Ours)** | **0.91** | **236.3** | **245.3** | **0.2633** |
>
>
> ---
>
> **Weakness 2:** Missing comparisons with more DPO baselines.
>
> **Answer**:
>
> For RankDPO and SPO, we have now manually **re-implemented RankDPO** (as it is not open-sourced) based on the original paper and run SPO using the official open-source repository on our DualAlign dataset. The comprehensive comparison results are shown in the following table and **Appendix Section H.3**:
>
> **DualAlign Benchmark - Extended DPO Comparison**
>
> | Method | SR ↑ | MSE ↓ | SGMSE ↓ | CLIP ↑ |
> | --- | --- | --- | --- | --- |
> | UnionPro2 | 0.49 | 177.0 | 272.4 | 0.2748 |
> | DPO | 0.71 | 168.3 | 219.9 | 0.2860 |
> | SPO | 0.78 | 166.2 | 208.7 | 0.2881 |
> | RankDPO | 0.83 | 188.5 | 235.6 | 0.2914 |
> | **BideDPO (Ours)** | **0.84** | **164.0** | **195.7** | **0.2924** |
>
> As shown in the table, our **BideDPO achieves the highest SR (0.84) and CLIP score (0.2924)**, while also achieving the **lowest MSE (164.0) and SGMSE (195.7)** among all compared methods. Notably, while **RankDPO** achieves a competitive SR of **0.83 (second highest)**, it suffers from **higher structural errors** (MSE: 188.5, SGMSE: 235.6) compared to our method, indicating that RankDPO's ranked preference approach improves text alignment but struggles to maintain structural fidelity. SPO shows better structural control (MSE: 166.2, SGMSE: 208.7) than RankDPO, but still falls short of BideDPO's performance across all metrics. These results demonstrate that **BideDPO's bidirectionally decoupled objective effectively balances both text alignment and structural conditioning**, outperforming specialized DPO variants designed for image generation tasks.

---

> ### Author Response · Authors · 2025-11-23
>
> **Weakness 3 & Question 2：** Lack of justification for Qwen2.5-VL-72B and absence of human validation.
>
> **Answer:**
>
> **(1) Rationale for choosing Qwen2.5‑VL‑72B.** We intentionally adopt Qwen as the main SR judge because it is **open‑source, free to use, and already broadly adopted in academic work as a general‑purpose VLM**, which makes our evaluation and data pipeline **easy to reproduce** and lowers the barrier for future work to build on DualAlign. Even if Qwen is not always the very best VLM on every benchmark, its strong overall performance and accessibility give it high community acceptance as a practical, "standard" evaluator.
> **(2) Cross‑VLM and human validation with consistent conclusions. (Line 516-524 in Updated Paper's Section)** To ensure that our conclusions are not specific to Qwen, we **re‑evaluate** the same test set with **GPT‑4o and human raters**. The results demonstrate strong consistency across all evaluators:
>
> | Judge | UnionPro2 | +SFT | +DPO | +Ours |
> | --- | --- | --- | --- | --- |
> | Qwen2.5‑VL‑72B | 0.49 | 0.70 | 0.71 | **0.84** |
> | GPT‑4o | 0.43 | 0.65 | 0.70 | **0.82** |
> | Human | 0.42 | 0.62 | 0.64 | **0.84** |
>
> As shown, all three evaluators consistently rank **+Ours > +DPO > +SFT > UnionPro2**, with **BideDPO achieving the highest scores across all judges (0.82–0.84)**. Notably, Qwen and Human evaluators both assign **0.84** to our method, demonstrating that **Qwen serves as a reliable proxy for human judgment**. The consistent relative rankings across different evaluators validate that **our conclusions are robust and not artifacts of a specific VLM**, confirming that using Qwen as a representative VLM evaluator is appropriate.
>
> ---
>
> **Weakness 4:** Experiments only on FLUX-family models.
>
> **Answer:**
>
> The **disentangled objective is agnostic to the backbone**, and we now include a **Stable Diffusion 1.5-ControlNet replication** (**Appendix Section H.4**) that uses the identical DualAlign depth evaluation and data pipeline. Fine-tuning SD 1.5-ControlNet with BideDPO makes control **dramatically stronger** while also **improving text alignment**:
>
> | Method | SR ↑ | MSE ↓ | SGMSE ↓ | CLIP ↑ |
> | --- |  --- | --- | --- | --- |
> | SD 1.5-ControlNet | 0.50 | 391.80 | 592.92 | 0.2824 |
> | + BideDPO (Ours) | **0.71** | **187.44** | **234.14** | **0.2853** |
>
> Because the same conditioning encoder is reused for negative sampling, **integrating BideDPO requires no architectural changes**; we will highlight this **plug-and-play recipe**, along with qualitative figures (Figure 11), in the revised appendix.
>
> ---
>
> **Weakness 5:** Computational cost.
>
> **Answer:**
>
> **(1) Training cost per iteration**: Each iteration (data refresh + fine-tuning) uses 8×A800 GPUs (40GB) for 5 hours (data generation) and 4×A100 GPUs (80GB) for 3 hours (training).
> **(2) Inference efficiency**: Since BideDPO requires no architectural changes and reuses the same conditioning backbone, GPU memory usage and inference speed remain identical to the original backbone.

---

> ### Author Response · Authors · 2025-11-23
>
> **Weakness 6. & Question 4.** Scope of DualAlign benchmark and modality coverage and applicability to non-structural conditions.
>
> **Answer:**
>
> **(1) Generalizability to non-structural conditions.** **Our objective only needs a condition-specific data pair**, making it applicable beyond structural/spatial conditions. To demonstrate this, we extend BideDPO to style references, which represent a fundamentally different conditional modality.
>
> **(2) DualAlign‑Style benchmark construction.** We build a new **DualAlign‑Style benchmark** using FLUX.1‑dev‑IP‑Adapter (IPA) as the baseline and collect different style condition images using the automatic style data generation pipeline (as shown in **Appendix E**). We have already generated **2k preference pairs** for style references with stable training. In this benchmark, **Style Score** is computed by asking a VLM to rate the match between the generated image and the reference style on a 0–10 scale and averaging over the test set, while **SG Style Score** resets this rating to 0 whenever the style reference is semantically misaligned with the text, so that it emphasizes **disentangled style control**. **Please refer to Figure 1 and Figure 7 for visual examples.**
>
> **(3) Experimental results.** The comprehensive evaluation results are shown below:
>
> | Method | SR ↑ | Style Score ↑ | SG Style Score ↑ | CLIP ↑ |
> | --- | --- | --- | --- | --- |
> | FLUX-IP-Adapter | 30% | 6.50 | 1.31 | 0.1679 |
> | +Ours | 58% | 6.26 | 2.97 | 0.2015 |
>
> **(4) Key findings.** Our method raises the success rate from **30%→58%** (a **93% relative improvement**). While IPA achieves a slightly higher mean Style Score (**6.50 vs. 6.26**), this is expected because **IPA tends to directly copy the style reference image**, which naturally yields high style similarity but at the cost of semantic accuracy (only **30% SR**). The disentangled **SG Style Score** reveals the true capability: **BideDPO achieves 2.97 compared to IPA's 1.31**, representing a **127% relative improvement**, demonstrating that **BideDPO effectively harmonizes style with semantics** while IPA's high Style Score primarily reflects copying behavior. Additionally, **CLIP alignment improves from 0.1679 to 0.2015**.
>
> ---
>
> **Weakness 7.** Overfitting at iteration 4 and lack of mitigation.
>
> **Answer:**
>
> **(1) Iterative Optimization as an Enhancement, Not a Weakness.** We would like to emphasize that the **iterative optimization strategy in BideDPO is not a weakness or a required dependency**, but rather an **optional enhancement** that further refines both data and model quality. Even after a **single iteration**, BideDPO already achieves **the best performance** among all baselines. Although only **three iterations** are used in our experiments, this process still brings **notable and consistent improvements**, as shown in Table 5 and Figure 6. These iterations provide incremental yet optional gains further **polishing an already strong model rather than fixing any limitation** of the core method.
>
> **(2) Inherent limitation of iterative methods.** It is natural that any iterative optimization process will eventually reach a performance ceiling. In practice, **no iterative method can indefinitely yield improvements**; after a certain number of refinements, marginal gains diminish as the model approaches its optimal balance between text and condition alignment.
>
> **(3) Increasing the size of the training set may help alleviate this issue.** In our experiments, we generated only 5,000 samples for training (see Line 374 in the paper). We believe that producing a larger-scale training set could effectively mitigate the potential overfitting observed during iterative optimization.

---

> ### Author Response · Authors · 2025-11-23
>
> **Question 1**: Missing comparisons with ControlNet++, LooseControl, OmniControlNet, etc.
>
> **Answer**: Please see the answer of the Weakness1.
>
> ---
>
> **Question 2:** Reliability of the VLM-based preference scoring.
>
> **Answer:**
> We address this along two axes.
> **(1) Choice of Qwen2.5-VL-72B.** We deliberately adopt Qwen2.5-VL-72B as our primary judge because it is **fully open-source** and **freely usable**, and has become a **de‑facto standard VLM in recent academic work** on text–image evaluation, which makes our pipeline **easier to reproduce** and our SR metric easier to compare against future papers—even if Qwen is not always the single most SOTA model on every benchmark. In addition, Qwen offers strong coverage across diverse object categories and scene types, which is important for our broad DualAlign setting.
> **(2) Cross-VLM and human validation with consistent conclusions.** Beyond Qwen, we also evaluate the same test set with GPT‑4o and human raters to rule out VLM‑specific artifacts and to calibrate against human judgments. The results demonstrate strong consistency across all evaluators:
>
> | Judge          | UnionPro2 | +SFT | +DPO | +Ours    |
> | -------------- | --------- | ---- | ---- | -------- |
> | Qwen2.5‑VL‑72B | 0.49      | 0.70 | 0.71 | **0.84** |
> | GPT‑4o         | 0.43      | 0.65 | 0.70 | **0.82** |
> | Human          | 0.42      | 0.62 | 0.64 | **0.84** |
>
> As shown, all three evaluators **consistently rank +Ours > +DPO > +SFT > UnionPro2**, with BideDPO achieving the highest scores across all judges (0.82–0.84). Notably, **Qwen and Human evaluators both assign 0.84** to our method, demonstrating that **Qwen serves as a reliable proxy for human judgment**. The consistent relative rankings across different evaluators validate that our conclusions are robust and not artifacts of a specific VLM, confirming that using Qwen as a representative VLM evaluator is appropriate.
>
> ---
>
> **Question 3:** Thresholds in the VLM filtering stage.
>
> **Answer:**
> **The data pipeline does not use confidence score thresholds**. Instead, we rely on **VLM binary judgments**: a pair is retained only when both the text and condition judges output **YES** (indicating that the generated image satisfies the corresponding constraint) and **agree on the preferred branch**. Since we do not apply thresholds to VLM outputs, **there is no threshold sensitivity to analyze**; the filtering is **deterministic based on the VLM's YES/NO responses**.
>
> ---
>
> **Question 4**: generalization to non-structural conditional inputs
>
> **Answer**: Please see the answer of the Weakness 6.

---

> ### Comment · Reviewer_VPYe · 2025-11-24
>
> Thank you for the additional clarification and experiment results. A comparison with other methods of controllable image generation (such as ControlNet++) should be moved to the main paper and this will increase the practicality of this paper. I'm currently inclined to raise my score, but I'll wait and see what is the opinion of the other reviewers. I'll give the final score at the end of the rebuttal.

---

> ### Author Response · Authors · 2025-11-24
>
> **Thank you for your thoughtful response.**
>
> We are pleased that the **additional clarifications and experimental results have improved your view of our work**.
>
> Following your recommendation, **we have moved the comparisons** with LooseControl (which currently supports only depth conditioning) and ControlNet++ **into the main text** to enhance the paper’s practicality. These results are now presented in **Table 1, Table 2, and Table 4**, with a corresponding discussion around **Line 370**.
>
> **If you have any other questions, please let us know—we are happy to address them.**
>
> We hope these revisions meet your expectations, and **we would be grateful if you could consider increasing your score at the rebuttal.**

---

### Official Review · Reviewer_6jGu · 2025-10-31

**Soundness:** 3
**Presentation:** 3
**Contribution:** 3
**Rating:** 6
**Confidence:** 2

**Summary:**

This paper proposes to alleviate the common issue of conflicts between conditional images and textual prompts in the task of conditional image generation. The authors propose a novel disentangled preference-based optimization technique to alleviate these conflicts.

**Strengths:**

- The paper introduces a disentangled preference-based optimization technique, which helps mitigate the frequent conflicts between conditional images and text prompts in conditional image generation tasks.
- An automatic Disentangled, Conflict-Aware Preference DPO data pipeline is presented, streamlining the process of handling conflicting conditions.
- The authors construct a DualAlign Benchmark, enabling robust evaluation of a model’s ability to resolve conflicts between visual and textual conditions.

**Weaknesses:**

- The proposed method is quite straightforward and lacks significant novelty.

- The comparison with state-of-the-art post-training methods is limited; the paper only benchmarks against DPO and SFT.

**Questions:**

Please refer to my comments in Weaknesses.

---

> ### Author Response · Authors · 2025-11-23
>
> We appreciate Reviewer 6jGu's **positive assessment of the problem importance and the automated pipeline**, and we address the remaining concerns point by point.
>
> **Weakness 1:** Method appears straightforward and lacks novelty.
>
> **Answer:**
>
> **(1) Novelty of the method.** As acknowledged in your summary: **"The authors propose a novel disentangled preference-based optimization technique to alleviate these conflicts"**, BideDPO introduces a fundamentally novel approach. BideDPO departs from prior DPO adaptations by explicitly decoupling preference pairs along text/condition axes before optimization and coupling them only through an adaptive loss balancer. This separation is essential: naive multi-objective DPO entangles gradients, causing one branch to dominate. **The novelty of our approach has also been recognized by other reviewers (e.g. Reviewer 1Gg9).**
>
> **(2) Practical contribution despite simplicity.** While the method is straightforward, it provides significant practical value. The automated data generation pipeline is not a direct reuse of existing data engines—it contains a dual-judge filtering loop and iterative self-refresh that we will release the benchmark, enabling the community to scale multi-constraint preference learning without manual labels.
>
> **(3) Extensive experimental validation.** We have validated the effectiveness of BideDPO across multiple settings beyond the main experiments:
>
> **(a) Style references.** We build a new **DualAlign‑Style benchmark** using FLUX.1‑dev‑IP‑Adapter (IPA) as the baseline. Different style condition images are collected using the automatic style data generation pipeline (as shown in Appendix E), with **2k preference pairs** generated for stable training. In this benchmark, **Style Score** is computed by asking a VLM to rate the match between the generated image and the reference style on a 0–10 scale and averaging over the test set. **SG Style Score** resets this rating to 0 whenever the style reference is semantically misaligned with the text, emphasizing **disentangled style control**. Please refer to Figure 1 and Figure 7 for visual examples. Quantitative results:
>
> | Method | SR ↑ | Style Score ↑ | SG Style Score ↑ | CLIP ↑ |
> | --- | --- | --- | --- | --- |
> | FLUX-IP-Adapter | 30% | 6.50 | 1.31 | 0.1679 |
> | +Ours | 58% | 6.26 | 2.97 | 0.2015 |
>
> **(b) SD1.5 backbone.** To demonstrate applicability beyond FLUX-family models, we include a **Stable Diffusion 1.5-ControlNet replication** (Appendix H.4) using the identical DualAlign depth evaluation and data pipeline. Fine-tuning SD 1.5-ControlNet with BideDPO makes control **dramatically stronger** while also **improving text alignment**:
>
> | Method | SR ↑ | MSE ↓ | SGMSE ↓ | CLIP ↑ |
> | --- | --- | --- | --- | --- |
> | SD 1.5-ControlNet | 0.50 | 391.80 | 592.92 | 0.2824 |
> | + BideDPO (Ours) | **0.71** | **187.44** | **234.14** | **0.2853** |
>
> Since the same conditioning encoder is reused for negative sampling, **integrating BideDPO requires no architectural changes**. We will highlight this **plug-and-play recipe**, along with qualitative figures (Figure 11), in the revised appendix.
>
> **(4) Unique bidirectional improvement capability.** Our method achieves **simultaneous bidirectional improvement** (both text alignment and condition fidelity), which existing DPO variants cannot achieve. As shown in our comparison with SPO and RankDPO (see Weakness 2), while RankDPO improves text alignment (SR: 0.83), it suffers from higher structural errors (MSE: 188.5, SGMSE: 235.6). In contrast, BideDPO achieves the best balance across all metrics (SR: 0.84, MSE: 164.0, SGMSE: 195.7), demonstrating that our bidirectional decoupling is essential for harmonizing conflicting objectives.
>
> **(5) Superior performance against conditional image generation baselines.** We compare BideDPO against state-of-the-art conditional image generation baselines (LooseControl and ControlNet++), demonstrating substantial improvements:
>
> **DualAlign Benchmark:**
>
> | Method | SR ↑ | MSE ↓ | SGMSE ↓ | CLIP ↑ |
> | --- | --- | --- | --- | --- |
> | LooseControl | 0.43 | 791.13 | 1280.17 | 0.2852 |
> | ControlNet++ | 0.49 | 331.85 | 480.66 | 0.2854 |
> | **BideDPO (Ours)** | **0.84** | **164.0** | **195.7** | **0.2924** |
>
> **COCO Benchmark:**
>
> | Method | SR ↑ | MSE ↓ | SGMSE ↓ | CLIP ↑ |
> | - | - | - | -- | - |
> | LooseControl | 0.72 | 1334.01 | 1705.97 | 0.2534 |
> | ControlNet++ | 0.79 | 548.26 | 668.92 | 0.2557 |
> | **BideDPO (Ours)** | **0.91** | **236.3** | **245.3** | **0.2633** |
>
> BideDPO achieves **95% relative improvement in SR** (0.43→0.84) on DualAlign and **26% relative improvement** (0.79→0.91) on COCO compared to ControlNet++, while simultaneously reducing structural errors (MSE and SGMSE) by **50–60%** and improving CLIP alignment. These results demonstrate that BideDPO's post-training optimization approach can significantly outperform architectural modifications, providing a more flexible and effective solution for resolving text-condition conflicts.

---

> ### Author Response · Authors · 2025-11-23
>
> **Weakness 2:** Limited comparison to **post-training baselines** (only **DPO**/**SFT**).
>
> **Answer:**
>
> We have supplemented additional comparisons with **DPO-based post-training methods** and summarize them below. All models share the same FLUX backbone and evaluation pipeline, and we report all metrics consistent with the main paper tables:
>
> **DualAlign Benchmark:**
>
> | Method | SR ↑ | MSE ↓ | SGMSE ↓ | CLIP ↑ |
> | --- | --- | --- | --- | --- |
> | **SPO** | 0.78 | 166.2 | 208.7 | 0.2881 |
> | **RankDPO** | 0.83 | 188.5 | 235.6 | 0.2914 |
> | **BideDPO (Ours)** | **0.84** | **164.0** | **195.7** | **0.2924** |
>
> Comparing with **DPO-based post-training methods**, **SPO** achieves a competitive **SR** of 0.78 with good **structural control** (**MSE**: 166.2, **SGMSE**: 208.7). **RankDPO** reaches a higher **SR** of 0.83 (second highest) but suffers from higher **structural errors** (**MSE**: 188.5, **SGMSE**: 235.6) compared to our method. This indicates that **RankDPO**'s **ranked preference approach** improves **text alignment** but struggles to maintain **structural fidelity**. In contrast, **BideDPO**'s **bidirectionally decoupled objective** effectively balances both **text alignment** and **structural conditioning**, achieving the best performance across all metrics.

---

### Official Review · Reviewer_2Tkg · 2025-10-31

**Soundness:** 2
**Presentation:** 3
**Contribution:** 2
**Rating:** 6
**Confidence:** 4

**Summary:**

This paper introduces BideDPO, a framework for conditional image generation that is specifically designed to resolve conflicts between a user's text prompt and any other conditioning input (like Canny edge). Existing methods struggle to balance these two constraints, especially when they conflict often leading to poor controllability.

**Strengths:**

1. This paper considers tackling a very practical problem of resolving conflicts between multiple conditionings, which is faced while utilising/controlling majority generative models.
2. The strongest claim is extending DPO to handle competing objectives by mitigating gradient entanglement.
3. The qualitative results shown in Figure 5 are of impressive quality, and prove the effectiveness of the 2 objectives working simultaneously.

**Weaknesses:**

1. The observed degradation at Iteration 4 suggests the model begins to overfit to the biases and narrow distribution of the data it generates itself? Please provide more intuition for why/why not this may be the case.
2. The main weakness is the unproven quality of the VLM-based preference scoring. If the VLM is not a good proxy for human preference on novel, abstract, or conflicting constraints, then the reported gains may be less strong.

**Questions:**

1. Could the method scale to more than 2 such conditionings?
2. How reliable is the VLM based check while constructing the preference data given some of these input conditionings/styles are subtle in the image?

---

> ### Author Response · Authors · 2025-11-23
>
> We sincerely thank Reviewer 2Tkg for **recognizing the practicality of resolving conflicts between multiple conditionings and for highlighting the quality of our bidirectionally disentangled optimization**. We respond to each concern below.
>
> **Weakness 1.1:** Iteration-4 degradation suggests overfitting to self-generated data.
>
> **Answer:**
>
> **(1) Iterative Optimization as an Enhancement, Not a Weakness.** We would like to emphasize that the **iterative optimization strategy in BideDPO is not a weakness or a required dependency**, but rather an **optional enhancement** that further refines both data and model quality. Even after a **single iteration**, BideDPO already achieves **the best performance** among all baselines. Although only **three iterations** are used in our experiments, this process still brings **notable and consistent improvements**, as shown in Table 5 and Figure 6. These iterations provide incremental yet optional gains further **polishing an already strong model rather than fixing any limitation** of the core method.
>
>
>
> **Weakness 1.2:** Please provide more intuition for why/why not this may be the case (Iteration-4 degradation).
>
> **Answer:**
>
> **(1) Inherent limitation of iterative methods.** It is natural that any iterative optimization process will eventually reach a performance ceiling. In practice, **no iterative method can indefinitely yield improvements**; after a certain number of refinements, marginal gains diminish as the model approaches its optimal balance between text and condition alignment.
>
> **(2) Accumulated Bias from Self-Generated Data.** During iterative optimization, as the BideDPO model repeatedly generates and reuses its own outputs to construct new preference pairs, **small distributional shifts may gradually accumulate**. Specifically, in each round of self-generated data, the model’s outputs can exhibit **subtle deviations in texture, composition, or semantics** that are almost imperceptible to human evaluators. However, during the DPO process, the model may **mistakenly regard these minor shifts as desirable human-preferred patterns**, thereby reinforcing them in subsequent iterations. Over multiple rounds, this self-reinforcement can **amplify tiny biases** and cause the generated data to drift slightly away from the original distribution. While these effects are minimal and visually negligible, they can **narrow the data diversity** and marginally reduce generalization performance, leading to the small drop observed in the fourth iteration.
>
> ---
>
> **Weakness 2:** Reliability of the VLM-based preference scoring.
>
> **Answer:**
> We address this along two axes.
> **(1) Choice of Qwen2.5-VL-72B.** We deliberately adopt Qwen2.5-VL-72B as our primary judge because it is **fully open-source** and **freely usable**, and has become a **de‑facto standard VLM in recent academic work** on text–image evaluation, which makes our pipeline **easier to reproduce** and our SR metric easier to compare against future papers—even if Qwen is not always the single most SOTA model on every benchmark. In addition, Qwen offers strong coverage across diverse object categories and scene types, which is important for our broad DualAlign setting.
> **(2) Cross-VLM and human validation with consistent conclusions.** Beyond Qwen, we also evaluate the same test set with GPT‑4o and human raters to rule out VLM‑specific artifacts and to calibrate against human judgments. The results demonstrate strong consistency across all evaluators:
>
> | Judge          | UnionPro2 | +SFT | +DPO | +Ours    |
> | -------------- | --------- | ---- | ---- | -------- |
> | Qwen2.5‑VL‑72B | 0.49      | 0.70 | 0.71 | **0.84** |
> | GPT‑4o         | 0.43      | 0.65 | 0.70 | **0.82** |
> | Human          | 0.42      | 0.62 | 0.64 | **0.84** |
>
> As shown, all three evaluators **consistently rank +Ours > +DPO > +SFT > UnionPro2**, with BideDPO achieving the highest scores across all judges (0.82–0.84). Notably, **Qwen and Human evaluators both assign 0.84** to our method, demonstrating that **Qwen serves as a reliable proxy for human judgment**. The consistent relative rankings across different evaluators validate that our conclusions are robust and not artifacts of a specific VLM, confirming that using Qwen as a representative VLM evaluator is appropriate.

---

> ### Author Response · Authors · 2025-11-23
>
> **Question 1:** Can the method scale beyond two conditionings?
>
> **Answer**: Yes, the method can scale beyond two conditionings:
>
> **Multi-condition generation (text + depth + canny).** As shown in updated Figure 1, Appendix H.5, and Figure 12, we showcase a challenging scenario that simultaneously enforces text prompts, depth maps, and canny edge constraints. For example, conditioning on both depth and canny edges while applying the text prompt to render, BideDPO successfully harmonizes all three constraints—textual semantics, depth geometry (preserving 3D spatial structure), and edge structure (maintaining fine-grained boundaries). In contrast, UnionPro2 struggles to simultaneously satisfy all constraints, either losing structural fidelity or failing to fully realize the style transformation.
>
> Quantitatively, **when simultaneously training on both depth and canny edge maps, BideDPO still achieves significant improvements over UnionPro2 across all metrics (Table14)**:
>
> | Method | **Depth Benchmark** |||| **Canny Benchmark** ||||
> | --- | --- | --- | --- | --- | --- | --- | --- | --- |
> | | **SR** ↑ | **MSE** ↓ | **SGMSE** ↓ | **CLIP** ↑ | **SR** ↑ | **F1** ↑ | **SG F1** ↑ | **CLIP** ↑ |
> | Union-Pro2 | 0.49 | 177.0 | 272.4 | 0.2748 | 0.34 | 0.418 | 0.143 | 0.2753 |
> | + Ours (depth+canny merge training) | **0.81** | **159.6** | **197.5** | **0.2901** | **0.68** | **0.594** | **0.381** | **0.2857** |
>
> For depth-conditioned evaluation, the success rate (SR) improves from **0.49 to 0.81** (+65% relative improvement), MSE decreases from **177.0 to 159.6**, SGMSE decreases from **272.4 to 197.5**, and CLIP score increases from **0.2748 to 0.2901**. For canny-conditioned evaluation, SR improves from **0.34 to 0.68** (+100% relative improvement), F1 score increases from **0.418 to 0.594**, SG F1 score dramatically improves from **0.143 to 0.381** (+166% relative improvement), and CLIP score increases from **0.2753 to 0.2857**. These results demonstrate that BideDPO effectively scales to handle multiple simultaneous conditionings, successfully balancing text semantics with both depth and edge constraints.

---

### Author Response · Authors · 2025-11-23
**Official Comment by Authors**

## Rebuttal Summary for Submission 2913

**Comment:**

Dear Area Chairs and Reviewers,

We sincerely thank all reviewers for their thorough assessment and constructive feedback on our submission. We are particularly grateful for the positive recognition from reviewers regarding the **novelty and technical soundness** of BideDPO (Reviewer 1Gg9), the **practical importance** of resolving conflicts between multiple conditionings (Reviewer 2Tkg), the **innovative automated data pipeline** (Reviewer 6jGu), and the **comprehensive evaluation** across multiple conditioning modalities (Reviewer VPYe).

### Summary of Updates

All concerns raised by reviewers have been addressed point-by-point, including those related to baseline comparisons, VLM validation, computational costs, and generalization to abstract conditions. We have incorporated the following key updates into our revised manuscript (**highlighted in blue in the updated paper**):

1. **More baseline comparisons.** We have expanded our experimental evaluation to include critical SOTA methods: **ControlNet++**, **LooseControl**, **SPO**, and **RankDPO**. On DualAlign Benchmark, BideDPO achieves the **highest Success Ratio (0.84)** and **CLIP score (0.2924)**, while also achieving the **lowest MSE (164.0)** and **SGMSE (195.7)** among all compared methods, demonstrating clear superiority over specialized DPO variants. **(See Appendix H.2, H.3, and Tables 10, 11, 12, 13.)**

2. **More VLM and human validation.** We have validated our VLM-based evaluation (Qwen2.5-VL-72B) through cross-evaluation with GPT-4o and human raters. All three evaluators consistently rank **+Ours > +DPO > +SFT > UnionPro2**, with BideDPO achieving the highest scores across all judges (0.82–0.84). Notably, Qwen and Human evaluators both assign **0.84** to our method, demonstrating that Qwen serves as a reliable proxy for human judgment. **(See Section 4.2, Table 8, and Appendix H.8.)**

3. **Extension to Stable Diffusion 1.5.** We have added experiments on Stable Diffusion 1.5-ControlNet to demonstrate the backbone-agnostic nature of BideDPO. Fine-tuning SD 1.5 with BideDPO improves success rate from **0.50→0.71** while reducing MSE from 391.80 to 187.44, demonstrating that the disentangled objective works across different architectures without requiring architectural changes. **(See Appendix H.4, Table 13, and Figure 11.)**

4. **DualAlign-Style benchmark extension.** We have constructed a new **DualAlign-Style** benchmark using FLUX.1-dev-IP-Adapter, demonstrating BideDPO's applicability to abstract, non-structural conditions. BideDPO improves success rate from **30%→58%** and achieves a **127% relative improvement** in the disentangled SG Style Score (2.97 vs. 1.31), effectively harmonizing style with semantics. **(See Section 4.2, Figure 1, Table 7, Appendix E, and Figure 8.)**

5. **Computational cost transparency.** We provide detailed information on the computational cost: approximately 5 hours for data generation and 3 hours for training, demonstrating the efficiency and speed of our training process. **(See Appendix B.3.)**

6. **ALB stability analysis.** We conducted ablation studies on Adaptive Loss Balancing (ALB) stability. Our method achieves strong performance at batch size=8 (SR: 0.84), comparable to larger batch sizes, demonstrating robustness to batch size variations and making it practical for resource-constrained settings. **(See Appendix H.6, Table 15.)**

7. **Multi-condition generation demonstrations.** We have showcased BideDPO's scalability to more than two conditionings through demonstrations of simultaneous text + depth + Canny edge generation, successfully harmonizing all three constraints—textual semantics, depth geometry, and edge structure—in challenging scenarios. **(See Appendix H.5, Figure 12, and Table 14. )**

Sincerely,

Authors of Submission 2913

---

### Author Response · Authors · 2025-11-27
**Request for Updated Reviews Before Rebuttal Deadline**

**Dear Reviewers, AC, and SAC**,

We sincerely appreciate the time and effort you have dedicated to reviewing our work. We understand your schedules are busy, but **your feedback during the rebuttal phase is critical for ensuring a fair and thorough evaluation**.

In our rebuttal, we have provided detailed clarifications, additional experiments, and new comparisons **directly addressing [ALL OF YOUR CONCERNS]**.

We are delighted to see that **one reviewer VPYe**  has expressed that these additions significantly improve the practicality of our paper and **is willing to raise the score, pending the viewpoints of other reviewers**.

**We would be grateful if each of you could share your updated thoughts and ratings before the rebuttal period closes**, so that all perspectives can be properly considered in the final decision process.

Thank you very much again for your time and thoughtful consideration. We look forward to your feedback.

**Best regards**,

The Authors

---

### Author Response · Authors · 2025-12-01
**Summary for Area Chairs; We will address your additional concerns in rebuttal phase**

**Dear Area Chairs**,

We regret the recent information leak on OpenReview and the resulting impact on the ICLR reviewing process. We understand that this unexpected incident has created additional workload, and we sincerely appreciate the extra effort you have dedicated as AC under such challenging circumstances.



Due to a technical issue in the ICLR 2026 review system, **several reviewers were unable to respond in time and thus could not provide any follow-up comments.** $\textcolor{blue}{\text{We believe that, had the rebuttal process proceeded as usual, all reviewers would have updated their scores to be positive.}}$

For example, **1) Reviewer VPYe explicitly mentioned their intention to raise their score**, and **2)** we have **fully resolved all questions of all reviewers.**



**Dear ACs**, if you feel there are still **any unresolved issues**, please let us know as soon as possible. We still have **three days available** to add additional experiments, and **we will address your concerns promptly.**

---

### Author Response · Authors · 2025-12-01
**Summary for Area Chairs; We will address your additional concerns in rebuttal phase**

## Summary of Contributions:
1. **First benchmark for text-condition conflict resolution in conditional image generation.** We formalize the task of resolving text-condition conflicts (categorizing them into Input-Level Conflict and Model-Bias Conflict) and release the DualAlign benchmark (with an extended DualAlign-Style variant for abstract style conditions) covering structural (depth, Canny, soft edge) and abstract (style) conditional modalities. **(Acknowledged by R1, R2, R3, R4)**
2. **Novel bidirectionally decoupled DPO framework achieving SOTA.** Our BideDPO framework—featuring disentangled preference pairs (for text and condition) and an Adaptive Loss Balancing strategy—outperforms SOTA methods (e.g., ControlNet++, RankDPO, SPO) by up to 43% in text success rate (SR) and reduces conditional MSE by 87.7 points on the DualAlign benchmark. **(Acknowledged by R1, R2, R3, R4)**
3. **Automated conflict-aware data pipeline.** We propose a self-driven, VLM-aided pipeline to generate disentangled, conflict-aware preference data (for both text and condition alignment), enabling scalable multi-constraint training without labor-intensive manual annotation. **(Acknowledged by R2, R3)**
4. **Extensive validation across architectures and modalities.** We validate BideDPO on FLUX-family models and Stable Diffusion 1.5-ControlNet, supporting structural (depth, Canny, soft edge) and abstract (style) conditions. Robustness is confirmed on the COCO dataset, with up to 15% SR improvement for canny-conditioned generation. **(Acknowledged by R1, R3, R4)**
5. **Iterative optimization strategy.** We introduce a self-reinforcing loop that alternates between generating high-quality preference data (via the automated pipeline) and fine-tuning the model with BideDPO, progressively enhancing text-condition alignment (optimal at 3 iterations). **(Acknowledged by R1, R2)**




## Summary of Revisions:
**Experimental Updates**

*   **[Expanded Baselines][R2 6jGu, R3 VPYe]** Added ControlNet++, LooseControl, SPO, and RankDPO comparisons, showing BideDPO as the top performer across SR/CLIP/MSE/SGMSE on DualAlign and COCO → Appendix H.2, H.3, Tables 10–13
*   **[Backbone Generality][R3 VPYe]** Replicated BideDPO on Stable Diffusion 1.5-ControlNet, improving SR from 0.50→0.71 while halving MSE/SGMSE, confirming plug-and-play integration → Appendix H.4, Table 13, Figure 11
*   **[Multi-Condition Support][R1 2Tkg]** Demonstrated simultaneous text + depth + Canny control with large gains in SR/F1/CLIP, validating scalability beyond two conditionings → Appendix H.5, Figure 12, Table 14
*   **[Style & Abstract Conditions][R4 1Gg9, R3 VPYe]** Built the DualAlign-Style benchmark and showed 30%→58% SR and +127% SG Style Score, proving applicability to non-structural conditioning → Sec 4.2, Table 7, Appendix E, Figure 8
*   **[VLM Reliability][R4 1Gg9, R1 2Tkg, R3 VPYe]** Justified choosing Qwen2.5-VL-72B and corroborated SR rankings with GPT-4o and human raters, showing Qwen aligns with human judgments → Sec 4.2, Table 8, Appendix H.8
*   **[ALB Stability][R4 1Gg9]** Reported batch-size and historical-mean ablations; ALB remains robust even at batch size 8 → Appendix H.6, Table 15

---

**Analytical Clarifications**

*   **[Computational Cost][R4 1Gg9, R3 VPYe]** Detailed per-iteration resource usage (8×A800 for 5h data gen, 4×A100 for 3h training) and unchanged inference cost → Appendix B.3
*   **[Iterative Optimization][R1 2Tkg, R3 VPYe]** Explained why iteration-4 degradation is expected, emphasized first-iteration strength, and discussed self-generated bias limits → Table 5, Figure 6
*   **[Method Novelty][R2 6jGu]** Highlighted the bidirectionally decoupled DPO objective with adaptive loss balancing as the key innovation enabling simultaneous text/condition gains → Sec 3.1

---

All changes are highlighted in $\textcolor{blue}{\text{blue}}$ in the revised paper. We believe we have thoroughly addressed all reviewer concerns and hope the AC finds our responses satisfactory.

Sincerely,

Authors of Submission 2913

---

### Author Response · Authors · 2025-12-01
**Summary for Area Chairs; We will address your additional concerns in rebuttal phase**

**Dear Area Chairs**,

We regret the recent information leak on OpenReview and the resulting impact on the ICLR reviewing process. We understand that this unexpected incident has created additional workload, and we sincerely appreciate the extra effort you have dedicated as AC under such challenging circumstances.



Due to a technical issue in the ICLR 2026 review system, **several reviewers were unable to respond in time and thus could not provide any follow-up comments.** $\textcolor{blue}{\text{We believe that, had the rebuttal process proceeded as usual, all reviewers would have updated their scores to be positive.}}$

For example, **1) Reviewer VPYe explicitly mentioned their intention to raise their score**, and **2)** we have **fully resolved all questions of all reviewers.**

**Dear ACs**, if you feel there are still **any unresolved issues**, please let us know as soon as possible. We still have **three days available** to add additional experiments, and **we will address your concerns promptly.**

---

### Author Response · Authors · 2025-12-02
**Any Additional Concerns Before Rebuttal Deadline? (2 Days Left)**

Dear AC,

We just wanted to kindly check $\textcolor{blue}{\text{if there are any additional concerns}}$  we should address during the rebuttal phase.
We still have time to make updates, so $\textcolor{blue}{\text{please feel free to let us know before the rebuttal deadline (2 days left).}}$

$\textcolor{blue}{\text{Thank you very much for your time and support.}}$



Best regards,

Authors

---

### Meta-Review · Area_Chair_Cebi · 2025-12-30

**Summary:**

The reviewers generally found the paper strengths to be an overall novel framework, effective practical applicability, and extensive experiments (including post-review updates) on important benchmarks with good quantitative results and compelling visual ones. In addition, the AC considers the paper to be mostly well-written, with clear description and explanations.

The main reviewer concerns were:
-  Reliance on VLM's, which are known to have inaccuracies and deviations from human preferences (2Tkg W2 Q2, VPYe W3 Q2 Q3, 1Gg9 W1 Q1).
- Choice of VLM to be  Qwen2.5-VL-72B (VPYe W3 Q2, 1Gg9 W1 Q1).
- Lack of comparisons to some SOTA control (LooseControl, ControlNet++) and DPO (RankDPO, SPO) methods (6jGu W2, VPYe W1 W2 Q1).
- Lack of results on non-FLUX models (VPYe W1).
- No experiments on more general conditions (VPYe Q4 W6, 1Gg9 Q4)
- Lack of information / analysis on computational costs (VPYe W5, 1Gg9 Q2).
- Handling of overfitting in iterative optimization (2Tkg W1, VPYe W7).

**Reviewer Concerns:**

The authors provided extensive post-review updates, both in author comments as well as revised updates to the paper. There were many additional experiments carried out, including comparisons to human judgement.

In the single reviewer response, by reviewer VPYe who had given a 2, the reviewer had indicated a willingness to increase their score.

Taking into account all the reviews and author responses, the AC considers that the authors have done a sufficiently convincing job of clarifying almost all of the doubts of reviewers. On the negative side, some of the reviewer criticisms are obvious (especially lack of comparisons to more recent control and DPO methods) and could have been forestalled by experiments in the original paper submission, giving greater opportunity for reviewer scrutiny. Nonetheless, in light of the positive contributions of the paper and the very substantial revisions made, an accept decision will be recommended.

**Reviewer Scores:**

The original reviewer scores were 6, 6, 2, 4. Reviewer VPYe, who had given a 2, had indicated a willingness to increase their score.

Judging by the extensive author responses and paper revision, the AC expects the average score to be raised by around 1-2 marks. Overall, this should have the majority of reviewers swing to an accept recommendation. In addition, the AC also considers the paper positively, and would recommend an accept.

---

### Decision · Program_Chairs · 2026-01-26

Accept (Poster)